Article 

# Environmental metagenomics enhances detection of circulating viruses from live poultry markets in Cambodia

Peter Cronin[1], Jurre Y. Siegers[2], Vireak Heang[2], Songha Tok[2], Sarath Sin[2], Ben Sievers[2], Victor Omondi[2], Sithun Nuon[2], Kimtuo Chhel[2], Janin Nouhin[2], Vutha Chim[3], Bunnary Seng[3], Makara Hak[4], Sorn San[5], Sothyra Tum[3], Filip F. Claes[6], Cadhla Firth[7], Yvonne C. F. Su[1], Gavin J. D. Smith[1] ✉ & Erik A. Karlsson[2] ✉

Environmental surveillance has emerged as a pivotal strategy for early detection of pathogens that pose a threat to humans. In Asia, live-bird markets (LBMs) are key human-animal interfaces for zoonotic virus transmission. Traditional sampling strategies are time-consuming, expensive and carry significant biosafety risks. Here, we assess the performance of metagenomics on environmental samples (ES) versus traditional poultry swabs for detecting viral pathogens in two Cambodian LBMs between January 2022 and April 2023. ES, including air (n = 35), cage swabs (n = 17), carcass wash water (n = 17) and drinking water (n = 9) are collected alongside oropharyngeal and cloacal swabs from chickens (n = 30) and ducks (n = 29). ES is sensitive in detecting 40 viruses from pathogen families including *Orthomyxoviridae* and *Coronaviridae*. Air samples capture the greatest diversity of poultry viruses. Viral contigs from ES show high sequence identity to poultry swab contigs when aligned to the same gene. We show ES outperforms poultry samples in detecting the highly pathogenic influenza A/H5N1, including clades 2.3.4.4b and 2.3.2.1c, which are found in the environment but are missed by poultry swabs. Our findings show metagenomics on ES replicates traditional surveillance, offering broader coverage and improved pathogen detection. This approach could be pivotal for mitigating zoonotic spillover and enhancing pandemic preparedness.

Environmental surveillance has become an increasingly important strategy for early detection and monitoring of viral pathogens[1–5]. Several major disease outbreaks in humans have resulted from zoonotic spillover from animals[6–11] making pathogen surveillance at human-animal interfaces such as live-bird markets (LBMs) critical.

LBMs are common in Asia, holding critical socio-economic significance. However, they are also marked by the prevalence of both endemic and emerging infectious diseases of One Health importance, including highly pathogenic avian influenza A virus (AIV) H5N1[2,12,13]. Highly pathogenic A/H5N1 has a mortality rate exceeding 50% in

¹Programme in Emerging Infectious Diseases, Duke-National University of Singapore Medical School, Singapore, Singapore. ²Virology Unit, Institute Pasteur du Cambodge, Pasteur Network, Phnom Penh, Cambodia. ³National Animal Health and Production Institute, Phnom Penh, Cambodia. ⁴Food and Agriculture Organization of the United Nations, Emergency Center for Transboundary Animal Diseases, Country Office, Phnom Penh, Cambodia. ⁵General Directorate of Animal Health and Production, Ministry of Agriculture, Forestry, and Fisheries, Phnom Penh, Cambodia. ⁶Food and Agriculture Organization of the United Nations, Emergency Center for Transboundary Animal Diseases, Regional Office for Asia Pacific, Bangkok, Thailand. ⁷College of Public Health, Medical and Veterinary Sciences, James Cook University, Townsville, QLD, Australia. ✉e-mail: gavin.smith@duke-nus.edu.sg; ekarlsson@pasteur-kh.org

humans[14,15] while also imposing a significant economic burden on global poultry industries[16,17]. Outbreaks of pathogens like A/H5N1 in poultry often lead to severe economic losses affecting livelihoods reliant on small scale poultry production[18–20]. Other pathogenic avian viruses, including members of the *Rotavirus*[21,22], *Avastrovirus*[23], *Gammacoronavirus*[24] and *Alpharetrovirus* genera[25] also cause substantial losses in domestic poultry. Improving pathogen surveillance systems at LBMs is essential to infectious disease monitoring, risk mitigation, and outbreak response, especially in developing regions of the world.

Classical surveillance and pathogen monitoring at LBMs has traditionally relied on random testing of individual animals via oropharyngeal and cloacal swabs. However, this approach may miss circulating pathogens if the selected bird is not infected when sampled, thus limiting the probability of detection. In addition, traditional sampling strategies are time-consuming, costly, and pose significant occupational biosafety risks while compromising animal welfare[26,27]. ES, particularly air, can provide a more comprehensive, safe, and easier approach to pathogen surveillance. Furthermore, it is hypothesized that poultry virus shedding can result in contamination of the surrounding environment, with the environment becoming a reservoir of pathogens exacerbated by insufficient biosecurity measures at LBMs[28,29]. Previously, we demonstrated through air sampling that workers at an LBM in Cambodia were exposed to influenza subtypes A/H5N1 and A/H9N2[30] and this approach has also proven successful in accurately detecting pathogens in other settings[31–34]. In addition, ES can be used to detect influenza A virus subtypes from sources such as the wash water used to clean bird carcasses after de-feathering at LBMs[30].

We hypothesize that ES could supplement traditional poultry pathogen surveillance and provide insight into risks of environmental exposure for LBM workers and consumers. This information is crucial for mitigating risks of pathogen transmission between birds and throughout the poultry supply chain, and for preventing spillover to humans. However, it remains unknown whether metagenomic data from ES in LBMs accurately reflects data from traditional poultry samples used for pathogen surveillance.

To date, environmental pathogen surveillance has largely relied on PCR-based methods, often used for monitoring of select human pathogens (e.g., influenza A and SARS-CoV-2) in wastewater and sewage[35,36]. This conventional approach for pathogen monitoring has been widely used in Cambodian LBMs for a number of years with a focus on avian influenza subtypes[37–42]. Although these approaches offer high sensitivity and specificity, they are limited in the range of viruses they can detect. Unbiased metagenomics has emerged as a method with a much broader target range for virus identification and surveillance, allowing detection of a diverse array of known and novel pathogens[43]. One major challenge using metagenomics is the low amount of viral nucleic acid present in most sample types, compared to higher levels of host and environmental nucleic material. Methods to reduce non-target sequences in metagenomic data have been explored, but vary in effectiveness[44]. However, viral enrichment using probe hybridization methods has been shown to significantly improve sensitivity across different sample types, enabling the accurate detection of low abundance and/or low-frequency virus variants[45–48].

In this study, we compared the performance of targeted virus capture probe-based metagenomics on several different ES types versus traditional poultry swab samples from LBMs for the detection and surveillance of avian pathogens. We hypothesize that using metagenomics on ES may offer a more robust approach to facilitate the early detection of virus transmission at high-risk human-animal interfaces, allowing for more rapid and effective risk mitigation measures to take place.

## Results

### Air detects more virus species than poultry swabs

To evaluate the performance of metagenomics for viral pathogen detection using ES compared to poultry swabs, we first calculated the detection rates of poultry viruses among both sample types. In total 84 different poultry virus species were identified over the course of the study (Fig. 1A). We showed that the majority of viruses found in poultry swabs could be directly recaptured in corresponding ES at the exact same timepoint with detection rates consistently ranging between 70% and 90% (Fig. 1A and Supplementary Fig. 3A, B). This finding was generally consistent across both locations and the majority of sampling dates when comparing the performance between individual ES and all four poultry swab types (Supplementary Fig. 4). Common poultry respiratory viruses including avian influenza and avian coronavirus were frequently identified in different poultry swab types and corresponding ES (Fig. 1A and Supplementary Fig. 5–8). While this was also true for other pathogens including Rous sarcoma virus, avian leukosis virus, avian orthoreovirus (*Retroviridae*) and chicken astrovirus (*Astroviridae*).

Despite the consistent overall performance of ES in capturing the same viruses found in poultry at each timepoint (70-90%), we observed variability in the performance of how individual ES types recaptured viruses specific to each of the four poultry swabs. By combining data from all twelve timepoints we show that air (slaughter and holding area) and cage swabs showed no significant differences in the number of viruses recaptured from chicken oropharyngeal and cloacal swabs (Supplementary Fig. 9 and Supplementary Data 6 and 7). Other ES including wash water, drinking water and air from outside the market recaptured significantly less viruses relative to both chicken sample types. Using this same approach, we show that all ES recaptured significantly less viruses than both duck swabs (Supplementary Fig. 10 and Supplementary Data 8 and 9). These findings highlight variation in performance across individual ES types, as well as differences depending on the poultry species targeted. Despite this, we found that most virus species directly identified in poultry were also detected across two or more ES types (Supplementary Fig. 11). However, at most timepoints, different ES groups uniquely detected poultry viruses that were not captured by any other ES group at that sampling date (Supplementary Fig. 11). Each ES group was implicated at least once in directly increasing the number of viruses recaptured. This suggests that independent of overall performance, a variety of sample types may be required to capture as many viruses as possible.

Viruses found in poultry swabs but failed to be recaptured in ES had a significantly lower abundance (Supplementary Fig. 12A) and were in circulation at the LBMs less often than those successfully detected through corresponding ES (Supplementary Fig. 12B). Importantly, there was no association between taxonomic family and the ability of ES to capture viruses identified in poultry swabs. For example, avian sarcoma virus CT10 could not be identified in ES despite appearing in poultry swabs on two separate occasions. However, ES successfully identified several other members of *Retroviridae* family including avian leukosis virus and rous sarcoma virus.

Despite variation in ES ability to directly recapture the exact same viruses found in poultry, it also frequently detected between 10 and 30 additional poultry viruses that were not identified through bird sampling at the same timepoint (Fig. 1A, Supplementary Fig. 5-10 and 13). Building on this we show that air (slaughter and holding area) and cage swabs had a significantly higher total virus diversity than all poultry samples (Fig. 1B, Supplementary Fig. 14 and Supplementary Data 10-13). In addition, cage swabs and air samples (from slaughter and holding areas) had significantly higher alpha-diversity than wash water and drinking water thus highlighting differences in ES performance (Fig. 1B). Using a similar approach we next quantified total virus diversity relative to each of the four poultry swabs (i.e., only including viruses that were detected at least

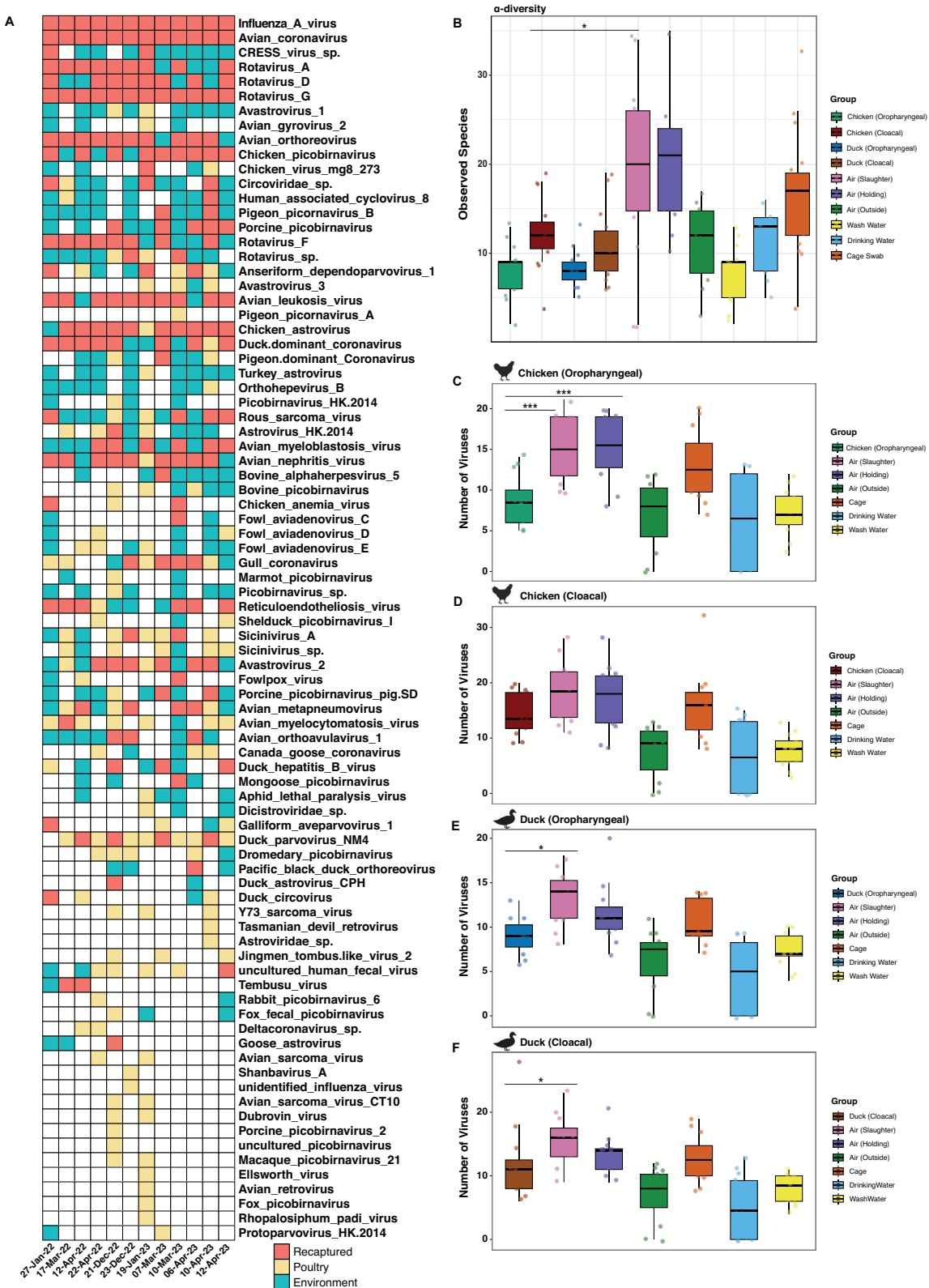

once in that specific poultry sample type). Air from the slaughter area detected significantly more viruses than duck samples and chicken oropharyngeal swabs but not cloacal swabs (Fig. 1C–F, and Supplementary Fig. 15 and Supplementary Data 14-17). Air collected from the holding area also captured more viruses than chicken oropharyngeal swabs, but no significant differences were observed when comparing to other bird samples.

ES analysed using targeted metagenomics revealed an additional 50 viruses which were not found through poultry sampling at any time during the study (and thus not included in previous analyses) (Supplementary Fig. 16). While most of these viruses may be from alternative sources (such as arboviruses or porcine viruses), 5 additional poultry viruses were also detected, including goose aviadenovirus A. duck adenovirus 4, pigeon circovirus, shelduck picobirnavirus II and

**Fig. 1 | Air detects more virus species than poultry swabs. A** Heatmap showing 84 different virus species which were detected at least once in poultry swabs over the course of the study. The red colour shows when a virus was identified in at least one poultry swab and at the exact same timepoint was recaptured in at least one environmental sample. The pale-yellow colour shows when a virus species was identified in poultry swabs but failed to be recaptured through environmental sampling. The blue colour shows virus species which were found in the environment but failed to be detected in poultry swabs. Rows of the heatmap represented virus species and columns represent individual environmental samples (annotated at the top of the heatmap). **B** Comparison of the alpha-diversity measure observed species between all ten groups included in this study ($P$ value shown = 0.02 and $n = 140$). Lastly the number of viruses detected in each environmental sample is compared relative to (**C**) chicken oropharyngeal ($n = 15$; $P = 0.004$ versus air slaughter area and $P = 0.007$ versus air holding area), **D** chicken cloacal ($n = 15$), **E** duck oropharyngeal ($n = 14$; $P$ value shown =0.01) and (**F**) duck cloacal ($n = 15$; $P$ value shown = 0.04) swabs. Comparative environmental samples were collected from air (holding area, $n = 13$; slaughter area, $n = 14$; outside, $n = 10$), wash water ($n = 19$) drinking water ($n = 9$), and cage swabs ($n = 17$). Each group is represented with data from independent biological samples. Boxplots display the median (centre line), interquartile range (box), and whiskers extending to 1.5× the interquartile range; points represent individual samples. Statistics were calculated using a Kruskal-Wallis with Dunns post-hoc test. All $P$ values obtained were corrected for false-discovery rate (FDR) using the Benjamini-Hochberg method. $P$ values are annotated as follows: $P < 0.05$ *; $P < 0.01$ **; $P < 0.001$ ***.

avian dependoparvovirus 1. Metagenomics on ES also detected the arbovirus dengue virus 3 as well as the common human respiratory pathogen *Rhinovirus A1* (Supplementary Fig. 16). This highlights a major advantage of ES in facilitating the detection of viruses which may be missed by traditional sampling methods or circulating pathogens from other sources.

Overall, these findings suggest that the ES conducted at LBMs in Cambodia enhances detection of poultry virus species at LBMs in Cambodia.

## Performance of environmental samples is dependent on poultry host species and sample type

Principal coordinate analysis (PCoA) of β-diversity (Jaccard distance) focusing only on the poultry viruses identified in this study (Fig. 1A) revealed statistically significant virome separation between the groups after controlling for location and time (PERMANOVA < 0.001*** and R2 = 0.037) (Fig. 2A). Pairwise PERMANOVA of each individual comparison revealed no significant differences between ES including cage swabs, drinking water and air from the holding area and slaughter area (Supplementary Data 18). Air samples from outside the LBM, as well as wash water, were significantly different from all other sample types. Both cloacal and oropharyngeal swabs collected from chickens had significantly different virome compositions compared to the same swab types from ducks (Supplementary Fig. 5). Given the observed differences in virome composition between poultry swab types we wanted to evaluate the relationship/performance with each sample type individually. Thus, based on the PCoA in Fig. 2A, we calculated the median coordinates (PCo1 and PCo2) (Fig. 2B) of each poultry swab (cloacal and oropharyngeal from both domestic ducks and chickens) and subsequently measured the distance of all samples from each point (Fig. 2C–F and Supplementary Data 19–22). Using this quantitative comparative evaluation of virome composition we show that no significant difference could be detected between oropharyngeal samples from chickens and ES of drinking water, cage swabs and air (slaughter and holding area) (Fig. 2C). Similar findings were observed when comparing cloacal swabs from chickens with drinking water, cage swabs and wash water, although air samples were a significantly larger distance by this measure (Fig. 2D). Interestingly, no difference could be detected between wash water and oropharyngeal or cloacal swabs from ducks when comparing their respective distances from each median centroid (Fig. 2E, F).

To better understand why specific viruses were not detected in ES, we first identified the poultry swab type most associated with taxa which failed to be captured in ES at the exact same they were identified in poultry. To do this, we calculated Spearman correlations between virus abundance (RPM) and the distance of each sample from poultry swab median centroid as calculated for Fig. 2B–F. The results of this analysis were then compared between poultry viruses successfully recaptured by ES and those which went undetected as shown in Fig. 1A (Fig. 2G–J). By employing this approach, we found that viruses which could not be detected through ES had significantly more positive correlations than viruses accurately recaptured when examining data for cloacal swabs in chickens (Fig. 2H). This result indicates that taxa undetected by ES tended to have a higher abundance the greater the distance from the median centroid thus highlighting a negative association with chicken cloacal swabs. Conversely, taxa successfully recaptured by ES tended to have more negative correlations than undetected taxa suggesting this group had a higher abundance the shorter the distance from the median centroid. This indicates that cloacal swabs from chickens are positively associated with taxa successfully recaptured. The opposite finding was true for oropharyngeal and cloacal swabs in ducks thus suggesting these sample types had a greater association with undetected taxa (Fig. 2I, J).

Using a Bayesian approach, we conducted source tracking analysis to estimate the proportion of viruses in ES that are likely to have come from each poultry swab type (Fig. 2K). We found that cloacal swabs from chickens were identified as the dominant source of virus composition for all ES. The probability of cloacal swabs from chickens contributing to the environmental virome composition was significantly higher compared to oropharyngeal swabs from chickens as well as both sample types from ducks (Fig. 2K). Typically, more chickens are processed at LBMs than ducks in Cambodia. This may in part explain our findings as the environment is contaminated with a greater number of chicken viruses.

Overall, these findings highlight the differential performance of ES with respect to poultry species and sample type. ES conducted at LBMs performs better at detecting viruses from chickens than those infecting ducks, while cloacal-derived chicken viruses appear to be a major source for the environmental pathogens in LBMs.

## Environmental contigs cover the majority of virus genes in poultry with a high sequence similarity

Next, we aimed to assess whether de novo assembled viral contigs from ES were an accurate representation of those found in poultry swab samples. At each independent timepoint, poultry contigs were aligned individually to the corresponding ES assemblies to calculate the percentage of poultry contigs recovered in ES. We found that the majority of contigs from direct poultry samples mapped to those from ES (Supplementary Fig. 17). For both oropharyngeal and cloacal swabs obtained from chickens, the highest percentage of viral contigs was shown to map to the ES air (slaughter and holding areas) and cage swabs (Supplementary Fig. 18 and Supplementary Data 23 and 24). Large variation in the percentage of viral contigs mapped was observed between air samples collected from outside the LBMs (Supplementary Fig. 18) highlighting substantial variability. No significant differences could be determined between the percentage of viral contigs mapped between ES and duck (Supplementary Fig. 19 and Supplementary Data 25 and 26).

To further assess the similarity between poultry and environmental sequences, all contigs were aligned to the full set of poultry virus genes which were detected in at least one poultry sample across the length of the study (see "methods" section). Applying the same methodology as the species-level analysis in Fig. 1, we demonstrate that most virus genes detected in poultry swabs were also recovered in

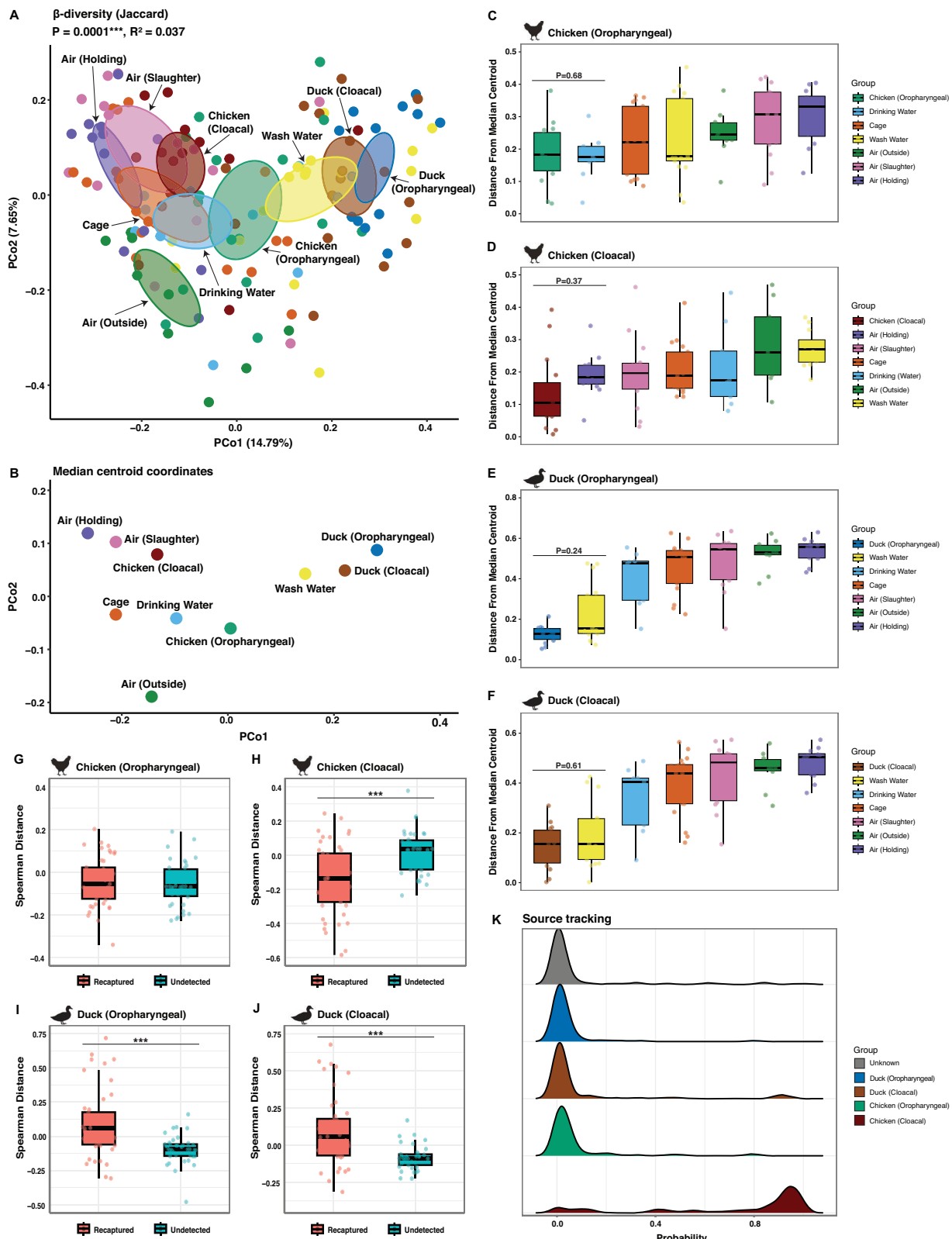

at least one corresponding environmental sample at the same time-point (Fig. 3A). The majority of genes identified belonged to species in a handful of highly abundant genera including *Coronaviridae*, *Orthomyxoviridae*, *Rotavirus* and *Orthoreovirus* (Supplementary Figs. 20–23). While ES broadly recapitulated the poultry swab virome with gene-level resolution we show marked variation in performance among individual ES types. Across all twelve sampling dates, air

collected from the slaughter area consistently recovered the highest number poultry virus genes (Supplementary Fig. 24) with no significant difference reported relative to each bird sample (Supplementary Fig. 25-26 and Supplementary Data 27-30). Cage swabs and air (holding area) performed similarly for chicken oropharyngeal and cloacal swabs but not duck samples (Supplementary Fig. 25-26). In contrast, carcass wash water and drinking water recovered markedly

**Fig. 2 | Performance of environmental samples (ES) is dependent on poultry host species and sample type. A** Principal coordinate analysis (PCoA) of beta-diversity (Jaccard) of poultry virus species. PERMANOVA was calculated adjusting for both location and time as a confounder. The eigen values are reported showing the variation reported by PCo1 and PCo2. **B** Based on the PCoA in A, we calculated the median centroid of each group by determining the median PCo1 and PCo2 coordinates. Relative to poultry swabs, we calculated the distance of all samples from (**C**) chicken oropharyngeal (*n* = 15), (**D**) chicken cloacal (*n* = 15), (**E**) duck oropharyngeal (*n* = 14), and (**F**) duck cloacal (*n* = 15) median centroids, as shown in panel B. Comparative environmental samples were collected from air (holding area, *n* = 13; slaughter area, *n* = 14; outside, *n* = 10), wash water (n = 19) drinking water (*n* = 9), and cage swabs (*n* = 17). Each group is ranked from the shortest to longest distance from its respective poultry median centroid. Spearman correlations between virus abundance in reads per million (RPM) and the distance from (**G**)

chicken oropharyngeal, **H** chicken cloacal (*P* value shown = 0.0003), **I** duck oropharyngeal (*P* value shown = 0.0001), and (**J**) duck cloacal (*P* value shown = 0.0001) median centroids were calculated for viruses either recaptured (*n* = 45) or undetected (*n* = 38) by environmental sampling, as shown in Fig. 1A. Each point represents an individual virus species, and therefore *n* values correspond to the number of distinct viruses rather than biological replicates. Boxplots display the median (centre line), interquartile range (box), and whiskers extending to 1.5× the interquartile range; points represent individual samples. **K** Ridge plot showing the probability of each poultry swab being the source of pathogenic viruses for all environmental samples. Statistics were calculated using a Kruskal-Wallis with Dunns post-hoc test. All *P* values obtained were corrected for false-discovery rate (FDR) using the Benjamini-Hochberg method. *P* values are annotated as follows: *P* < 0.05 *; *P* < 0.01 **; *P* < 0.001***.

fewer viral genes than any of the four poultry swab types (Supplementary Fig. 25-26). Although these findings persisted at each sampling date, carcass wash water uniquely captured a subset of genes that no other ES type detected on multiple different occasions, underscoring its surveillance value (Supplementary Fig. 27).

ES frequently recovered a substantial number of additional poultry virus genes often that were missed by concurrent bird sampling (Supplementary Fig. 28, 29). Aggregating data across all twelve sampling dates and restricting the analysis to viral genes relative to each bird sample individually we show that air (from slaughter and holding areas) and cage swabs exhibited significantly greater viral gene diversity compared to both duck swabs (Supplementary Fig. 30 and Supplementary Data 31, 32). Air from the slaughter area yielded a significantly higher number of detected viral genes compared to chicken cloacal swabs (Supplementary Fig. 31 and Supplementary Data 33-34). Notably, no other pairwise comparisons reached statistical significance, suggesting that wash water and drinking water performed comparably to poultry swabs in terms of total virus gene diversity (Supplementary Fig. 30, 31). Collectively, these findings further confirm that virus gene recovery varies by ES type and poultry host, underscoring the need for a multi-sample strategy to enhance surveillance.

To understand how accurate environmental sampling was in capturing the genetic makeup of poultry virus contigs, we next calculated pairwise percent identity (PID) and coverage for every poultry environment contig pair that aligned to the same gene. By calculating the difference in PID (poultry PID − ES PID) for contig pairs aligned to the same gene we show that most environmental contigs diverge from their poultry counterparts by ≤1%, indicating that the majority of sequences are near-identical (Fig. 3B–D). This finding was remarkably consistent when comparing each ES type to each poultry swab individually (Supplementary Fig. 32–35). Applying the same analysis to sequencing coverage revealed that, for contig pairs mapped to the same gene, coverage differences were minimal in most cases (Fig. 3E). Across all poultry swab types, air samples (slaughter and holding area) together with cage swabs exhibited the highest number of sequences which accurately paired to corresponding poultry contigs (Fig. 3F, Supplementary Fig. 36–38 and Supplementary Data 35–38). ES contigs with a corresponding poultry counterpart were primarily represented by species in the *Rotavirus* or *Coronaviridae* genus (Fig. 3F and Supplementary Fig. 39). Similarly, most paired poultry contigs belonged to the *Orthomyxoviridae* or *Coronaviridae* genus (Fig. 3F and Supplementary Fig. 39). Notably, a high proportion of poultry contigs which were not accurately represented by a corresponding environmental contig belonged to the *Orthomyxoviridae* family (A/Influenza) (Supplementary Fig. 40, 41). Closer inspection of each individual gene revealed the majority were of these genes were PB1, PB2 and PA for *Orthomyxoviridae* or ORF1ab for *Coronaviridae* (Fig. 3G). This finding may in part be explained by the virion having PB1, PB2 and PA

buried within the internal polymerase complex rendering them vulnerable to high levels of degradation once outside the host.

Overall, these findings highlight that ES enhances detection of poultry virus genes. De novo assembled contigs have a remarkably similar coverage and percent identity when aligned to the same gene further highlighting the accuracy of this approach.

### Environmental samples yielded highly pathogenic influenza A contigs more often than poultry swabs

Across the length of the sampling period, at both LBM locations, we identified four different influenza hemagglutinin (HA) subtypes in ducks and chickens including H5, H6, H7 and H9 (Fig. 4A, B and Supplementary Fig. 42, 43). H5 clade 2.3.4.4b was found on three separate visits to the market (Fig. 4A, C). However, this H5 clade was jointly detected by poultry swab sampling and ES on only one sampling date. By contrast, the same H5 clade 2.3.4.4b was detected in wash water, cage swabs, and air samples on two additional occasions without corresponding poultry detections (Fig. 4C). Similar findings occurred for clade 2.3.2.1c, which was detected at LBMs on four separate sampling visits (Fig. 4A, C). On one occasion (December 2022), both clade 2.3.2.1c and 2.3.4.4b were detected in eight different ES (4 air, 2 cage swabs and 2 wash water samples), suggesting high prevalence in birds coming into the market, and significant environmental contamination. However, no sequences were identified through direct animal sampling on this occasion, highlighting the versatility of environmental sampling. This event did not coincide with any festival period, even though earlier work in Cambodia has shown that increased poultry traffic at LBMs and increased H5N1 virus circulation are directly linked to major festival periods[49].

In total, HA subtype H6 was found in 27 samples across the study, with both the ST2853-like and HN57-like clades detected. The HN57-like clade was found in ducks but was not detected in corresponding ES (Supplementary Fig. 42), and this was the only detection of the HN57-like clade of H6 influenza in this study. On six separate sampling visits, the ST2853-like H6 clade was successfully sequenced from poultry swabs as well as air samples and carcass wash water (Supplementary Supplementary Fig. 42). On one visit, H6 was detected in air but not in the corresponding poultry swabs (Supplementary Fig. 4D). The Eurasian avian lineage of H7 HA was only found once and was sequenced from both air and poultry samples at the same time and place (Supplementary Fig. 43). H9 HA was the predominant HA subtype detected, with a total of 43 positive samples across all 12 visits (Fig. 4B). On seven different sampling visits, H9 was identified in both poultry samples and ES. Importantly, on four separate sampling visits, H9 sequences were found in data from ES but were not identified in data from poultry swabs (Fig. 4B). The opposite trend was also reported, whereby H9 sequences were detected in duck samples but were not observed in any ES. Overall, metagenomics using ES at LBMs detected circulating influenza HA subtypes more often than metagenomics using traditional poultry swabs.

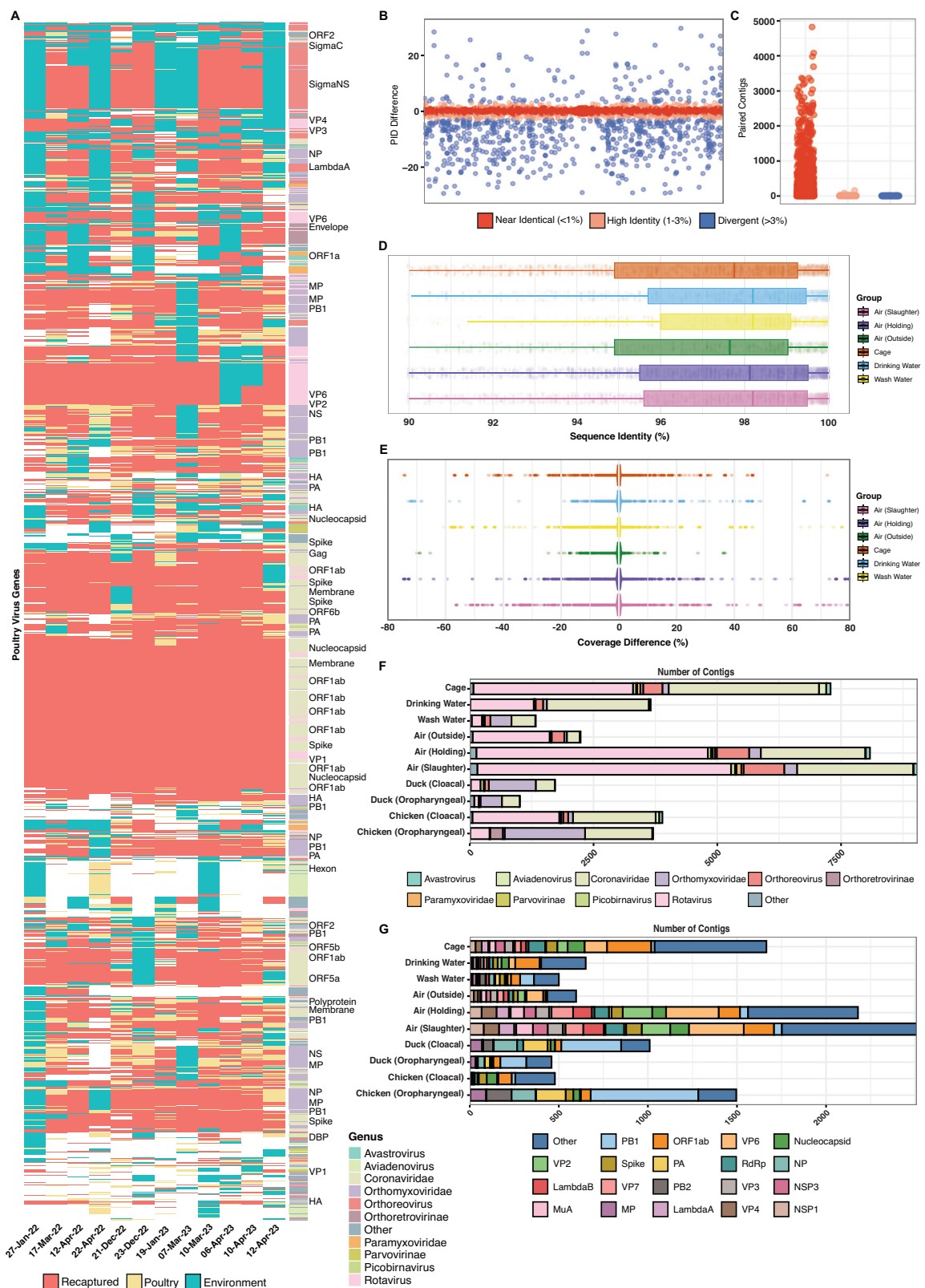

Three different neuraminidase (NA) gene subtypes were identified at LBMs in this study including N1, N2 and N6 (Fig. 4C–F). In line with H5 HA results, we identified both the 2.3.4.4b and the 2.3.2.1c clade of N1 (Fig. 4C). The 2.3.4.4b N1 clade was only detected in poultry swabs and ES (drinking water and air) on two visits (Fig. 4B, C). Six different ES yielded the detection of both H5 and N1 genes (representing the 2.3.4.4b and 2.3.2.1c clades) on December 23, 2022, at the LBM in

Takeo province. N2 was the most common NA subtype found throughout the study and was consistently co-detected in both poultry swabs and ES (Fig. 4B, D). In line with other findings, N6 sequences of the H5 Eurasian lineage were also consistently co-detected in both poultry samples and ES (Fig. 4B, F).

Hierarchal clustering based on influenza HA and NA subtype detection revealed that the HA H9 commonly co-occurred with N2 at

**Fig. 3 | Environmental contigs cover the majority of virus genes in poultry with a high sequence similarity. A** Presents a heatmap of detection rates for over 4,000 poultry virus reference genes (identified using DIAMOND blastx) in poultry swabs versus environmental samples. Red indicates that genes were found in both groups at the same timepoint, pale yellow represents detections only in poultry swabs, and blue for genes detected only in the environment. Rows correspond to individual virus genes annotated by genus (shown in the lower-right corner) and major clusters are labelled on the right. **B** For each gene (*x* axis) the difference in percent identity (PID) between poultry and environmental contigs is shown. **C** shows the number of poultry environment contig pairs at specified PID thresholds. **D** Distribution of percent identity (PID) values for environmental contigs aligning to the same reference genes as those detected in poultry. Boxplot displays the median (centre line), interquartile range (box), and whiskers extending to 1.5× the interquartile range; points represent individual samples. **E** Differences in percent coverage between poultry and environmental contigs are shown for each environmental sample type. Poultry samples comprised chicken oropharyngeal (*n* = 15), chicken cloacal (*n* = 15), duck oropharyngeal (*n* = 14), and duck cloacal (*n* = 15) swabs. Environmental samples were collected from air (holding area, *n* = 13; slaughter area, *n* = 14; outside, *n* = 10), wash water (*n* = 19), drinking water (*n* = 9), and cage swabs (*n* = 17). Each group represents independent biological samples. **F** For contig pairs aligned to the same gene(s), this plot displays the relative proportions of the major virus genera within each sample group. **G** For contigs that aligned uniquely to a single gene (unpaired), this plot shows the relative proportion contributed by each gene. Statistics were calculated using a Kruskal-Wallis with Dunns post-hoc test. All *P* values obtained were corrected for false-discovery rate (FDR) using the Benjamini-Hochberg method.

both LBMs (Fig. 4B). Using this approach, we were also able to infer that HA H6 (ST2853-like clade) co-occurred mostly with N6 while both H5 clades 2.3.2.1c and 2.3.4.4b co-occurred with N1 clades of the same subtype. While by nature metagenomic sequencing and de novo assembly of viral influenza A contigs from the environment and pooled poultry samples makes full length genome identification impossible for segmented RNA viruses, hierarchal clustering of detection patterns allows us to infer (co)-circulating subtypes.

## Discussion

Here we demonstrate that using targeted metagenomics on environmental surveillance samples can be an effective approach to enhance detection of viral sequences at LBMs. We found a clear association between viruses identified through poultry swabs and those detected in the surrounding environment. Our analyses revealed high sequence similarity between viral contigs assembled from both sample types. Notably, metagenomics using ES detected AIV H5 clades 2.3.2.1c and 2.3.4.4b more frequently than metagenomics using traditional swabs. The sampling and analysis methodologies described have the potential to enable faster responses to disease outbreaks at LBMs, offering a more efficient and cost-effective alternative to conventional pathogen surveillance, especially in high-risk settings such as LBMs.

Previous work on pandemic respiratory viruses, including influenza A viruses and SARS-CoV-2, has shown that these viruses can remain infectious for extended periods outside the host[50–52], surviving in various environmental conditions and mediums such as air[53–57], inanimate surfaces[53,58–60], and wastewater[36,61–63]. While similar environmental persistence has been reported for AIVs, limited information is available about other avian pathogens[64,65]. In this study, detection of multiple poultry pathogens from several viral families by metagenomics in the LBM environment demonstrates that ES is a useful method of sample collection in such settings. Furthermore, analysis of ES facilitated the detection of viruses that were not found in poultry swabs. While poultry may have been the host for some for these viruses, the majority of viruses may be from alternative sources/hosts yet still pose a significant threat to human and animal health. Importantly, this finding highlights the expanded surveillance coverage provided by ES, allowing the identification of pathogens which may otherwise go undetected.

Our findings show that poultry pathogen viral nucleic acids persist outside the host in LBMs, possibly increasing the risk of infection of naive poultry through indirect transmission routes. Airborne transmission emerged as a particularly concerning pathway, as air from slaughter and holding areas contained a higher diversity of poultry viruses than other ES. The performance of air in this study may be attributed, in part, to low background contamination. Air typically contains less bacterial and host material which results in cleaner libraries and more efficient sequencing. Moreover, air samples collected near food stalls adjacent to the poultry processing area also contained poultry viral sequences. These results imply that market employees, consumers, and bystanders may be at increased risk of infection via airborne transmission. These findings emphasize the argument for improved air quality and the physical separation of slaughter areas to mitigate the potential spread of poultry viruses. However, further studies are needed to determine if the nucleic acids detected in environmental samples represent viable, infectious viruses capable of replication.

Although each individual ES could not perfectly recapture all viruses found in poultry swabs at each sampling date, ES exhibited high specificity in detecting viruses which associated with different poultry sample types. For instance, chicken cloacal swabs were associated with air and cage swabs. Despite detecting the lowest number of pathogens, drinking water and carcass wash water associated with chicken oropharyngeal swabs and both duck sample types. Like drinking water, carcass wash water likely associated with oropharyngeal viruses due to the release of respiratory fluids after rinsing of the upper airway. The pooling effect after the washing of multiple carcasses results in a concentrated representation of respiratory virus in this sample type. However, differences in sample collection methods, matrices, and transport conditions may contribute to lower RNA yields or increased degradation in environmental samples compared to direct poultry swabs. This can influence downstream sequencing quality and may partially explain reduced virus detection in some environmental sample types. These results and limitations highlight the need for sampling diverse environmental types to maximize pathogen detection efficiency, particularly in markets where multiple bird species are present at different densities/quantities.

LBMs typically experience rapid turnover of poultry, with new batches arriving daily from multiple suppliers. Traditional random poultry sampling methods often fail to capture the full range of circulating viruses, as they rely on the assumption that sampled birds are infected. In contrast, our study shows that ES detects AIVs more frequently than traditional sampling. Despite the use of probe hybridization to enrich viral nucleic acids and improve sensitivity, some low-abundance taxa found in poultry swabs were undetected in environmental samples. These viruses were rare in the LBM during the study period, suggesting that low infection and replication rates may limit their environmental detection. Even with viral enrichment, some pathogens may fall below the detection threshold in environmental samples[66,67].

Undetected poultry pathogens were primarily associated with the oropharyngeal and cloacal virome of ducks, while ES was more effective at capturing chicken pathogens. This may in part be explained by a larger quantity of chickens (10-100 per day) being slaughtered than ducks (1-30 per day), at the particular LBMs sampled[68]. Another explanation for this may be variations in virus shedding patterns between species[69–71]. Previous meta-analysis has shown that chickens shed higher concentrations of certain influenza subtypes and other viruses through the cloaca compared to ducks[70]. Our study supports these findings, with the cloaca of chickens emerging as the predominant source of viral pathogens in ES. With respect to this, direct animal sampling may be particularly important for species present in

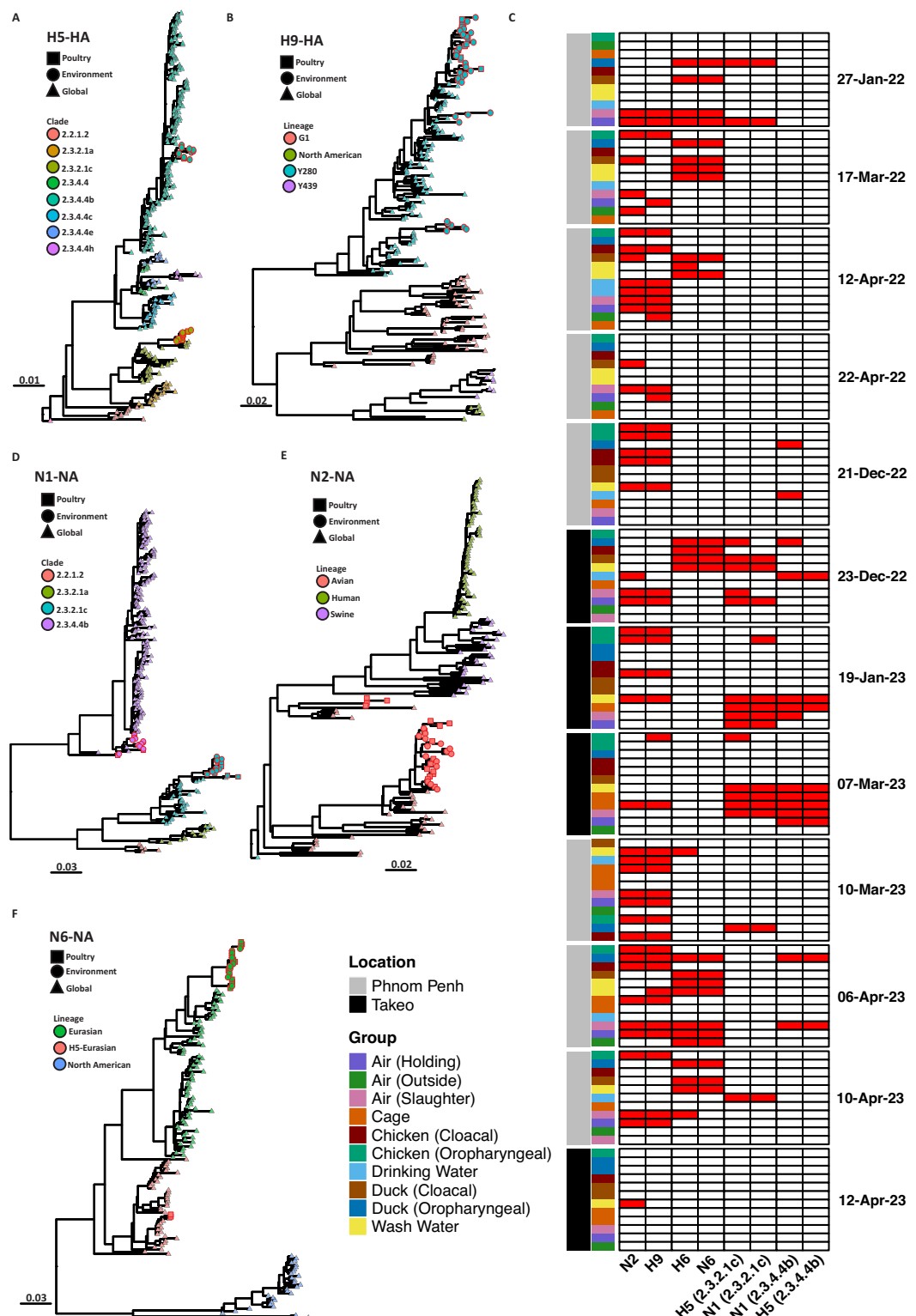

**Fig. 4 | Environmental sampling detected highly pathogenic influenza A contigs more often than poultry swabs. A, B** Maximum likelihood phylogenies of the hemagglutinin (HA) subtypes A) H5 and B) H9. **C** Heatmap showing the detection rate of different HA and neuraminidase (NA) subtypes across all twelve visits to the respective markets. Ward.d2 hierarchal clustering reveals HA and NA subtype combinations which co-circulated. Columns represent individual samples, annotated by location and sample type; rows represent HA and NA genes which were detected at LBMs. Red colour indicates that the influenza gene segment was detected, whereas white represents samples in which the segment failed to be detected. **D–F** Maximum likelihood phylogenies NA *Influenza* subtypes (**D**) N1, (**E**) N2, (**F**) N6. Each tip is coloured based on clade designation shown in the legend accompanying each phylogenetic tree.

lower numbers at human-animal interfaces to ensure their viral diversity is adequately captured. Therefore, ES enhancement of poultry swab metagenomics provides a more complete surveillance strategy, maximizing the likelihood of detecting diverse pathogens circulating in live poultry markets.

One limitation of this study is the potential assembly of chimeric contigs from ES. While our results show that all contigs have a high agreement (percent identity) with known poultry virus sequences publicly available on GenBank, the presence of chimeras assembled from different but closely related strains cannot be ruled out[72,73]. It should also be noted that virus annotations were based on the closest DIAMOND blastx match in the nr database, which does not always provide complete certainty of viral identity, particularly for short or partial contigs. Where the detected sequence divergence of key viruses could have pathogenic implications, targeted strain-specific PCR assays should be undertaken to independently verify the assembly-derived variants. Additionally, this current framework is unable to detect reassortment events in segmented viruses. Future work incorporating long-read sequencing or strain resolved assembly methods can help address these gaps. Given that phylogenetic trees in this study included sequences of varying lengths and genome coverage, phylogenetic inference may be influenced by non-overlapping regions among viruses, which could affect tree topology and branch length estimation.

Based on our findings, the optimal sampling strategy combines ES with strategic poultry swabbing to fully maximize virus detection in LBMs. ES of air and cage surfaces should be prioritized, as these sample types consistently detected the highest diversity of poultry viruses. At the same time, targeted oropharyngeal and cloacal swabs especially from ducks remain important for detecting species-specific viruses that may not persist in the environment. Together, this hybrid approach broadens detection while reducing biosafety risks, costs and dependence on animal handling. ES should not be viewed as a replacement for targeted animal testing, but rather as a complementary approach. ES does not allow for estimation of infection prevalence within individual animals or across market populations, limiting its utility for assessing transmission dynamics or quantifying risk at the population level. Environmental samples reflect pooled contamination from multiple species and individuals, preventing attribution to specific hosts or tracing to particular production systems. However, when used together with traditional surveillance approaches, combination strategies provide a more complete understanding of virus circulation and transmission risk, combining breadth of detection with the depth of host-specific investigation.

Nevertheless, environmental surveillance does enable timely and actionable risk mitigation strategies in high-risk settings such as LBMs. For instance, detection of high viral loads on cage swabs can prompt targeted disinfection protocols, while elevated contamination in air samples may inform interventions such as covering defeathering machines or improving ventilation systems. These mitigation measures can be implemented rapidly and at relatively low cost, without the need for individual animal testing. Follow-up environmental sampling can be used to assess the effectiveness of these interventions, helping to guide decisions on whether additional measures, such as temporary market closure or short-term poultry quarantine (e.g., 48–72 h), are warranted. Furthermore, the detection of novel or high-risk viruses through metagenomic analysis can trigger more targeted responses, including confirmatory PCR, host-specific sampling, or upstream supply chain tracing. Although environmental surveillance cannot localize infection to a specific source, it serves as a scalable early warning system that supports practical, real-time risk reduction in high-density, multi-species settings.

Overall, it is clear that ES, especially in high-risk interfaces like LBMs, can provide a snapshot of circulating viral diversity, including pathogens shed by multiple species, without the need to individually sample large numbers of animals. Instead of testing potentially thousands of birds to understand the full spectrum of viral circulation, a small number of strategically selected environmental samples can capture this diversity more efficiently. This pooled exposure effect not only increases the likelihood of detecting circulating and emerging viruses but also significantly reduces the logistical and financial burdens associated with large-scale animal sampling, RNA extraction, and sequencing. When combined with metagenomic approaches, environmental samples enable detection of viruses across multiple hosts and sample types, and reduce cost of molecular diagnostics, providing a powerful tool for early warning and outbreak preparedness. In addition, in this study, we used the CZ ID platform, an open-source and freely available metagenomic analysis tool designed to function with minimal computational infrastructure and limited bioinformatics expertise. Taken together, this approach enables more efficient and practical pathogen surveillance, making it especially well-suited for implementation in Southeast Asia and other low and middle-income countries, where scalable and sustainable surveillance solutions are urgently needed.

## Methods

All research conducted in this study complied with relevant ethical regulations. Sample collection protocols were approved by the National Ethics Committee for Health Research (NECHR) under the Ministry of Health, Cambodia (approval numbers NECHR013, NECHR143, NECHR149, and NECHR320, approved in 2021–2022). Cambodia does not have a dedicated animal ethics review board; however, all animal sample collections were carried out in collaboration with the National Animal Health and Production Institute and conducted with authorization from the General Directorate of Animal Health and Production, Ministry of Agriculture, Forestry and Fisheries.

### Study design and sample collection

Between January 2022 and April 2023 samples were collected on 12 occasions in Cambodia at Orussey market, Phnom Penh province and Daun Keo market, Takeo province (Supplementary Fig. 1). Orussey market was chosen as a sampling site for this study because is the largest LBM in Cambodia and serves as a major hub for poultry trade with birds arriving from a broad geographic area. This centralization of poultry trade increases the potential for pathogen transmission and amplification positioning Orussey market as a key site for surveillance. In contrast, Daun Keo market in Takeo province was selected as a study site due to its smaller scale, lower trade volume, and primarily regional bird sourcing which includes cross-border poultry trade. It is representative of the many small-scale, decentralized live bird markets dispersed throughout Cambodia. Individual animal samples consisted of oropharyngeal and cloacal swabs collected from ten domestic ducks and chickens. Birds were randomly selected from the entire population present at the market, ensuring that no two birds originated from the same cage/stall. Samples were combined into pools of ten individual cloacal or oropharyngeal swabs for sample processing, with ducks and chickens pooled separately. ES comprised air, carcass wash water, cage swabs, and drinking water. Air samples were collected for three hours at three locations at each market (the slaughter area, the poultry holding/storage area, and a "control area" - either the food storage area located in a separate building or > 50 m away from the LBM) using AerosolSense™ Samplers (Thermo Fisher Scientific). Air was sampled through an omnidirectional inlet at a rate of 200 L/min. Only one air sample from each location within the market was collected at each timepoint (Supplementary Data 1). Collected material was eluted from each air sampler by rinsing and vortexing the collection matrix in 2 mL of viral transport medium. Carcass wash water (50 mL) used to wash poultry carcasses after slaughter and de-feathering processes was collected from the slaughter area. Up to two wash water samples was collected per timepoint, as the slaughter area in each LBM typically has no more than two carcass wash containers (Supplementary Data 1).

Cage swabs were collected from five randomly selected cages in the poultry holding area of the market. All five cage swabs were subsequently pooled for sequencing. Where available, up to five 2 mL samples of poultry drinking water from inside poultry cages was collected at random (Supplementary Data 1). Samples were collected using polyester swabs and placed into viral transport medium (VTM) which was prepared in-house and consists of 2.95% tryptose phosphate broth, 145 mM NaCl, 5% gelatin, 54 mM Amphotericin B, 106 U/L penicillin-streptomycin, and 80 mg/L gentamicin. All samples were transported immediately and on ice to Institut Pasteur du Cambodge located in Phnom Penh. Samples were aliquoted into tubes on arrival and stored at −80 °C until further processing.

## RNA extraction, library preparation and next-generation sequencing

Samples were vortexed for 15–30 seconds before proceeding directly to RNA extraction using the Direct-zol RNA Miniprep Kit (Zymo Research, California, USA) according to the manufacturer's protocol and eluted in 50uL of nuclease-free water. We used 200 μL of sample per extraction. Swab samples were pooled by type and species: ten samples per pool for oropharyngeal and cloacal swabs from chickens and ducks, five samples per pool for cage swabs, up to two samples pooled for wash water samples and five samples pooled for drinking water. Sequencing libraries were generated using the Twist Library Preparation Enzymatic Fragmentation (EF) kit 2.0 (Twist Bioscience, San Francisco, California, USA) and the Twist Comprehensive Viral Research Panel (CVRP), according to the manufacturer's protocol (Twist Total Nucleic Acids Library Preparation EF Kit 2.0 for Viral Pathogen Detection and Characterization Protocol and Twist Target Enrichment Standard Hybridization v1 protocol). Briefly, extracted RNA was converted to cDNA using the ProtoScript II First Strand cDNA Synthesis kit (New England Biolabs, Ipswich, Massachusetts, USA) and Random Primer 6 (S1230S) from New England Biolabs (NEB). The NEB Next Ultra II Non-Directional RNA second Strand Synthesis kit (E6111S) was subsequently used to convert the single-stranded cDNA to double stranded DNA (dsDNA). Illumina TruSeq-compatible libraries were generated using the Twist Library Preparation EF kit 2.0. Library preparation included DNA fragmentation at 37 °C for 5 minutes, followed by 10 cycles of PCR amplification, and 22 cycles for post-capture PCR amplification. The pooled libraries were enriched by hybridization capture using the Twist Comprehensive Viral Research Panel (CVRP). Following enrichment, libraries were pooled in equimolar ratios, diluted and denatured prior to sequencing according to the standard MiSeq System Denature and Dilute Libraries Guide (Document # 15039740v10). The library was then sequenced to generate paired-end 75 bp reads using a MiSeq V3 reagent kit (Illumina, MS-102-3001) on the Illumina MiSeq System. In order to quantify potential contamination, nuclease-free water was included as a negative control and processed alongside both poultry and environmental samples from the point of extraction through to library preparation, and subsequently sequenced on the same Illumina sequencing run as the study samples. Sequencing results from the negative controls are presented in Supplementary Fig. 2. We demonstrate that negative controls contained few detectable viral contigs none of which was shared with other samples in this study. This effectively rules out any contamination and provides a meaningful baseline for interpreting true viral signals in study samples.

## Processing of viral sequencing data

Raw sequencing data was processed using the CZ ID pipeline (https://czid.org) which is an open-source cloud-based metagenomics pipeline for global pathogen detection and monitoring[74]. Briefly, the Illumina bioinformatics pipeline employed through CZ ID (https://chanzuckerberg.zendesk.com/hc/en-us/articles/360034790554-Illumina-Pipeline-Details) first quality filters and trims raw sequencing reads with fastp (Supplementary Data 2)[75]. Host-derived sequences were then excluded by aligning quality-controlled reads to the human, chicken, and duck reference genomes using Bowtie2[76]. The remaining non-host reads were assembled de novo into contigs using SPAdes[77]. Taxonomic classification of contigs was performed via DIAMOND blastx[78] against the NCBI nr protein database. Finally, reads were mapped back to the assembled contigs using Bowtie2 for abundance estimation. Viral abundance was estimated as reads per million (RPM) relative to the total number of reads which mapped to virus sequences, while de novo assembled contigs were directly downloaded from the CZ ID website. In order to remove potential false positives virus contigs were only retained for downstream analysis if they accounted for greater than 0.1% of reads in at least one sample.

## Bioinformatic analysis

All biostatistical analyses were carried out in Rstudio (version 4.4.0)[79]. Alpha-diversity was estimated using the estimate_richness function from the phyloseq package in R. Beta-diversity (Jaccard Index) was calculated using the vegdist function in the vegan package (version 2.7)[80]. To statistically test for differences between sample types we employed a Permutational Analysis of Variance (PERMANOVA) test which was calculated using the adonis function also in the vegan package. Pairwise PERMANOVAs were calculated using the pairwise.adonis function. To visualise virome composition between sample types we used principal coordinate analysis (PCoA) using the dudi.pco function of the ade4 package (version 1.7)[81]. To perform a quantitative comparative evaluation of virome composition we employed a median centroid testing methodology based on the Jaccard distances from a PCoA. This approach sees the median PCo1 and PCo2 coordinates calculated (median centroid) for each poultry swab sample type and the distance of all other samples from this point is subsequently determined. Spearman correlations were calculated using the corr.test function in the psych package[82]. In order to estimate the proportion of viruses in ES that are likely to have come from each poultry swab type source tracking analysis was conducted using the R package SourceTracker[83]. SourceTracker is a Bayesian approach that estimates the proportional contribution of potential source environments to a given sink sample, based on community composition. The model uses Gibbs sampling to infer the probability distributions of sources contributing to each sink, allowing for the inclusion of an "unknown" source category to account for uncharacterized variation. All plots used to visualise data were produced using the ggplot2 package in R[84]. Heatmaps were produced using either the ComplexHeatmap package[85] or the circlize package[86]. For visualisation using heatmaps, virus abundance in RPM was rank scaled using a custom function in R. For a given array of values 'x' of length k, denoted by $x = \{x_1, x_2,..., x_k\}$, the values were first ranked. The corresponding rank-scaled values, denoted by $x^R = \{x_1^R, x_2^R,..., x_k^R\}$, were then calculated using the formula: $x_i^R = [Rank(x_i) - Minimum(Rank(x))] / [Maximum(Rank(x_i)) - Minimum(Rank(x))]$. This transformation involves ranking the original values and rescaling those ranks to fall within the [0, 1] range. Statistical significance was determined using the non-parametric Wilcox test between two comparative groups. When statistical significance was being determined between three or more comparative groups the non-parametric Kruskal-Wallis with Dunns post-hoc test was employed using the PMCMRplus package. All P values obtained were corrected for false-discovery rate (FDR) using the Benjamini-Hochberg method (p.adjust function in R). P values are annotated as follows unless otherwise stated: $P < 0.05$ *; $P < 0.01$ **; $P < 0.001$***.

## Sequence identity and coverage

Contigs de novo assembled from poultry swabs were first mapped to those recovered from environmental samples with BWA-MEM2[87], and for each environmental sample we used Samtools[88] to calculate the proportion of poultry contigs that had at least one homologous environmental counterpart (number of mapped poultry contigs

divided by the total poultry contigs, expressed as a percentage). To extend the comparison to the gene level, we extracted the top DIAMOND blastx hit for every poultry contig after taxonomic assignment, retrieved the corresponding GenBank proteins, and compiled a custom reference set of ~4000 viral genes detected in at least one poultry sample (Supplementary Data 3). All quality-filtered contigs were then queried against this database with DIAMOND blastx; coverage, alignment length, number of mismatches and percent identity were recorded (Supplementary Data 4). Only alignments with an e-value $\leq 1 \times 10^{-5}$ were retained. When contigs from a poultry sample and its paired environmental sample both aligned to the same reference gene, their percent identity and coverage values were compared, whereas genes with no contig meeting the e-value threshold in each sample were recorded as undetected. Differences in percent identity (PID) were determined by comparing poultry and environmental contigs that aligned to the same reference gene, with the PID of the environmental contig subtracted from that of the poultry contig.

### Phylogenetic tree reconstruction

For phylogenetic tree reconstruction of influenza gene segments from de novo assembled contigs all available reference sequences were downloaded from the NCBI Influenza database and GISAID (Supplementary Data 5) from January 1, 2014 to July 1, 2024. From our metagenomic data, only influenza contigs de novo assembled through the CZ ID pipeline and greater than 200 bp were included for analysis. Contig taxonomy was determined using blast[89]. Reference datasets were subsampled to 500 sequences using SMOT[90] and combined with sequences identified in the current study. SeqKit was used for sequence length filtering and FASTA file manipulation[91]. Multiple nucleotide sequence alignments were carried out using Clustal Omega with default settings[92]. The resulting multiple sequence alignments were trimmed to ensure alignment accuracy using trimAl[89,93]. After trimming, alignments were manually inspected and any remaining erroneous or ambiguous regions were removed. Maximum likelihood phylogenies for each gene segment or genome were individually reconstructed using IQ-TREE (version 2.1.4)[94] with the best fit nucleotide substitution model. The phylogeny was inferred using 1000 bootstrap replicates, and branches were filtered based on bootstrap support. Specifically, terminal branches whose immediate parental node had support <70% were pruned using the *ape::drop.tip()* function in R. Internal branches with bootstrap support <70% were retained to maintain overall topological structure. Phylogenetic trees were plotted in R using the ggtree package[95].

### Reporting summary

Further information on research design is available in the Nature Portfolio Reporting Summary linked to this article.

## Data availability

Data used in this study is publicly available on the European Nucleotide Archive under accession number PRJEB83776. The Supplementary Datasets provide the GISAID accession numbers for all viral sequences analysed in this study; these sequences were obtained from publicly available repositories and were not generated by the authors.

## Code availability

All analysis scripts, processed data tables, and figure-generation code used in this study are available at Zenodo (https://doi.org/10.5281/zenodo.17597121) and mirrored on GitHub (https://github.com/PeterCroninX/air-sampling-zoonotic-virus).

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

## Acknowledgements

The investigators thank everyone involved in the critical discussions and review of this manuscript, as well as all those contributing to live bird market and influenza surveillance and response in the Kingdom of Cambodia. This includes teams at the National Institute for Public Health, the Cambodian Communicable Disease Control Department, Ministry of Health, the National Animal Health and Production Institute, General Directorate of Animal Health and Production, Ministry of Agriculture, Forestry, and Fisheries, the World Health Organization, the United States Centres for Disease Control and Prevention, provincial health and agriculture directors, market managers, middlemen, vendors, and sellers from live bird markets, and the Virology Unit at Institut Pasteur du Cambodge, all of whom contributed to this study. We gratefully acknowledge all data contributors, i.e., the Authors and their Originating laboratories responsible for obtaining the specimens, and their Submitting laboratories for generating the genetic sequence and metadata and sharing via the GISAID Initiative, on which this research is based. Research reported in this publication was supported by the Bill and Melinda Gates Foundation (INV-049293), the National Institute of Allergy and Infectious Diseases of the National Institutes of Health Centres for Research in Emerging Infectious Diseases under subaward #12-312-0217530 issued under prime award #1U01 AI151378-01, the Food and Agriculture Organization of the United Nations (FAO), contract 75N93021C00016 from the National Institute of Allergy and Infectious Diseases, NIH, Department of Health and Human Services, USA, and by the Duke-NUS Signature Research Programme funded by the Ministry of Health, Singapore. The text as published does not necessarily represent the official views of the US-CDC, WHO, FAO, or the National Institutes of Health. The content is solely the responsibility of the authors.

## Author contributions

P.C. performed bioinformatics analysis, data curation and validation, data visualisation and manuscript writing. J.Y.S. and V.H. conceptualised the study, performed data curation and reviewed/edited the manuscript. S.T., S.S., B.S V.O., S.N. and K.C. collected samples and performed next-generation sequencing preparation. V.C., B.S., M.H., S.S., S.T., F.C., and J.N. conceptualized the study. C.F., YC.F.S., G.J.D.S. and E.A.K. conceptualised the study and reviewed/edited the manuscript. All authors substantially contributed to the discussion of content.

## Competing interests

The authors declare no competing interests.
