## [Peer Review file · Nature Communications]

Environmental metagenomics enhances detection of circulating viruses from live poultry markets in Cambodia

Corresponding Author: Dr Erik Karlsson

Version 0:

Reviewer comments:

Reviewer #1

(Remarks to the Author)

Cronin et al. examined the utility of using environmental sampling at live bird markets for surveillance of influenza and other viral pathogens of concern. Zoonotic pathogens pose a serious threat to human health, and live animal markets are especially hazardous when it comes to zoonoses. Environmental sampling is less expensive and easier logistically than directly swabbing animals. They found that most of the viruses detected from direct swabs were found through environmental sampling, although there were differences in the viruses found in each environmental sample type. Environmental sampling was better at detecting the chicken virome than the duck virome. Environmental sampling is an important surveillance tool and it is critical to understand how well it performs compared to more traditional methods

Major Issues-

The usefulness and impact of this study would be greatly improved if recommendations on optimal environmental sampling were given. In lines 247-250 it states that "a variety of sample types are required to accurately capture the most viral viruses from poultry". It would be helpful if this was followed by how combining specific environmental sampling types increases the ability to recapture the virome from the direct swabs.

The title doesn't accurately reflect the conclusions. Air sampling alone isn't sufficient to detect the range of viruses and environmental sampling better reflects chicken than duck viromes. Something like "Environmental sampling closely recapitulates direct chicken swabs for avian viruses circulating in live poultry markets" would be better.

The github link doesn't currently work.

The figures overall are very nice, but there are a few issues that impede the ability to interpret them. All of the air samples are colored the same despite being treated as separate categories for analysis. This is an issue when not all data points are labeled, such as in Figure 1C, Supplementary Figure 4, and Supplementary Figure 7. The following issues are more minor. Within the figures the order of the samples isn't consistent and takes a little extra effort to read (i.e Figure 2 D-G and Figure 3H). For Figure 2A, the circle is unnecessary, and the figure would be easier to interpret if it was linear. Figure 4 – It would be easier to see what is going on if the tip shapes were a bit smaller or slightly transparent.

Minor issues

Line 298 -301: Exactly how many additional avian viruses were detected

Supplementary Figure 5 is labeled as Supplementary Figure 4.

Reviewer #2

(Remarks to the Author)

The authors present a new, robust and more inclusive approach to the surveillance of avian viral pathogens in high-risk settings, environmental metagenomic surveillance, which provides a better overview of the viral diversity in such settings due to its ability to detect several pathogens at the same time, using only minimal resources in the collection of environmental samples at the same time.

The authors demonstrate that the same pathogens can be detected accurately by metagenomics on environmental samples as by metagenomics on poultry samples. However, the authors also acknowledge that several different environmental sample types may need to be combined to achieve a similar level of coverage to poultry samples.

The introduction covers all the important aspects related to different surveillance strategies, but it could provide some more context about the specific study location. The proposed approach is methodologically and technically coherent and demonstrates the authors' expertise in the application of the described laboratory and (bio-)statistical methods. All methods are described in detail and reproducible. The results are underpinned by numerous useful illustrations. The discussion addresses all the key points raised in the text, although some limitations in the interpretation and follow-up of the results need to be discussed. For example, it remains unclear which specific intervention strategies can be chosen based on the results and where they should be applied. In addition, aim to prove the superiority of the proposed approach over "conventional pathogen surveillance", but it remains unclear what this baseline is.

Introduction:

-Could you provide a short summary of studies looking into avian viral pathogens in the study location by "conventional pathogen surveillance"?

-Could you please explain what your understanding of "conventional pathogen surveillance", to which you are comparing your approach, is? At the beginning of the discussion, you seem to compare your approach to "targeted probe-enriched metagenomics on direct poultry swab samples", but it does not become clear from the text before.

Methodology:

-Why were exactly those two live bird markets chosen?

-In some instances throat swabs are mentioned, in others oropharyngeal swabs. Which ones were used and why?

Oropharyngeal swabs would be the preferred choice for the detection of avian influenza viruses. Was viral transport medium used and what were the transport conditions on the way to the laboratory? All this may impact the probability of extracting sufficient high-quality RNA for sequencing.

Discussion:

-It would be useful to discuss how the use of different sample collection methods, sample types and transport conditions may have impacted amount and quality of the RNA extracted.

-Why in your opinion are air samples performing better than other environmental samples?

-Why do you think is the approach better at detecting chicken than duck viruses?

-Why do you think carcass wash water provided statistically significantly different results?

-The authors claim that the proposed approach is more cost-effective. Is there any evidence or an example of what kind of resources may be saved?

-Can the proposed approach realistically be implemented on a wider scale in Cambodia or similar settings? Do laboratories in such settings typically have the necessary equipment and expertise to carry out this type of analysis compared to, e.g., conventional PCR testing?

-How can the proposed approach complement rather than replace existing approaches?

-The authors claim that the proposed approach allows for more rapid and effective risk mitigation. Which specific mitigation strategies can be chosen based on the results and where should they be applied? It would be important to discuss here that the proposed approach is not able to identify the infected population within the setting, which would be necessary to design targeted mitigation strategies, e.g. for specific poultry production systems. Given that infection cannot be traced back to a certain population, intervention at the source is not possible. For example, avian influenza viruses are found in most live bird markets in Southeast Asia, so how would a positive test result be used?

Other comments:

-Is the title appropriate given that the authors are talking about more environmental samples than just air samples?

-The data, code and analysis files under <https://github.com/PeterCx/air-sampling-zoonotic-virus> are not available.

-Table 1 is not mentioned in the text.

-Lines 48-49: please check for some newer references on wastewater surveillance.

-Line 82: the results do not indicate at which point along the poultry supply chain interventions are appropriate.

-Lines 292-294: does this include pathogens not typically associated with the respective host (chickens or ducks)?

-Lines 341-344: maybe the season played a role? Or some sort of festivity?

Reviewer #3

(Remarks to the Author)

Overall comments:

This paper investigates approaches for virus surveillance in live bird markets. It describes the comparison of viruses detected in cloacal and throat swabs from poultry to those detected from environmental samples (air, carcass wash, cage swabs). Similar viruses were found in the environmental samples compared to the poultry swabs, including some viruses that were likely not from the targeted poultry species (chickens and ducks). This indicates that environmental surveillance approaches could be useful for conducting surveillance of viruses in live animal market settings.

Whilst use of environmental surveillance for zoonotic and veterinary pathogens is not new, this study provides a potentially powerful direct comparison of results from environmental samples and swabs. There are however currently some major

weaknesses in the study design or reporting. I detail these concerns below, but key areas include a severe lack of clarity and detail in reporting of methods (e.g., numbers of samples per time point), apparent absence of negative control samples, and phylogenetic estimation from virus contigs that have been created from environmental samples that likely represent chimeras of virus genomes from many different individuals. I am also concerned about some elements of statistical analysis, such as little effort to statistically control for differences in the number and type of samples per time point. I hope that the authors can address these concerns.

Major comments:

What appropriate blank negative controls were sequenced here (e.g., a blank swab taken in the market, an unused air filter)? Given that the aim is to compare different sample types, it would be extremely important here to confirm that the diversity of viruses observed are not the result of contamination between samples during sequencing. Metaviromic studies can also easily detect viruses present via other contamination in the lab or reagents, and this can be excluded with appropriate controls.

Line 111: The number of samples collected is not present in the methods. The information is roughly summarised in Table 1, but this is never referred to in the text. In any case, it surely needs to be more detailed, i.e., broken down per market, per time point, and per sample type and species. It's not clear how bird samples/stalls were selected – were these taken from both the slaughter and the poultry holding area as well? How many stalls? Without this information, it's impossible to evaluate whether environmental sampling is better, or whether it simply targets broader or different locations.

Line 157: The bioinformatic information is not sufficiently detailed. Please provide information on the mapping approaches (e.g., what is the reference database used here), as well as information on sequence length filtering etc (e.g., what parameters). What number of reads or contig length is considered a "detection", and how does this compare to negative controls?

Line 192: GISAID is mentioned and sequences are used in phylogenies. Yet there is no acknowledgements table of the submitting and originating labs, or other equivalent evidence of information to appropriately acknowledge the authors sharing data on this platform. This would be required according to the terms of use of GISAID. (To be clear, I recognise that NCBI data does not require such acknowledgement according to the terms of use).

Paragraphs line 152 and line 193: It is problematic to build contigs from environmental or pooled data and rely on this for phylogenetic tree estimation. This is because chimeras that represent similar infections in many different hosts can be easily generated. At a minimum, it would be needed to perform bootstraps and provide support for grouping of environmental and non-environmental samples, and provide some comment on the limitation of this approach in terms of possible inaccuracies in estimated divergence and the potential challenges of relying on chimera sequences for public and veterinary health surveillance.

Line 193: It's unclear whether the authors used alignments that included both full length CDS and also shorter sequences (200bp) when building phylogenies. This could deeply skew phylogenetic tree estimation and phylogenetic placement. It is possible that better handling of these data is implied by "manually curated and.... trimmed to ensure alignment accuracy". Please provide additional details of this.

Ethics: How were human reads screened in and removed from the metagenomic data, particularly the air sampling data, before submission to GenBank? Bait capture would not necessarily exclude these.

Figure 1: it's a shame that Figure 1A isn't broken down into different sample types. The difference between a carcass wash – which will presumably also pick up blood-borne pathogens – and an air sampler is quite profound. I'm not convinced that directly comparing throat/cloacal samples to the whole collection of these other sample types is very fair here, especially given the number of samples per group isn't known.

Figure S1 – some of these viruses have only 1 RPM. How many million reads were generated per sample? There must be some minimal cut-off of RPM below which these data cannot be trusted. What do the sequencing negative controls suggest? Avian viruses could presumably be from other birds, e.g., wild birds or domestic geese.

Discussion: I was surprised that the discussion didn't mention many limitations or weaknesses, especially compared to existing sampling approaches. A key limitation would surely be the inability to quantify prevalence per market. In addition, there would be an inability to detect reassortment segmented viruses in environmental samples, compared to individual birds, and also the fact that assembly could result in chimeric viruses. Assessing the implications of detected variants in such chimeric viruses would be more challenging.

Section line 311: What is the value of testing mapping of contig-to-contig here? Particularly when contigs are short and sparse, a contig from, e.g., a capsid found in a throat swab, would obviously not map to a contig of an RdRp found in an environmental sample. So is this proportionality more related to how much of the total genome was recovered in the environmental sample, rather than anything interesting about diversity? It would be useful to clarify this to the reader. If so, perhaps % of genome per type would be a better metric.

In general, there are statistical issues drawing conclusions about the ability of one sample type to capture another, when the number and type of samples is variable per time point. For example, Section line 330: Making such comparisons is challenging given the relatively limited information on sampling. For example, looking at Figure 3, no ducks were swabbed in Takeo in 23rd December 2022. It's perhaps unsurprising then that no H5 viruses were found in bird samples, because H5 viruses are highly duck-adapted rather than chicken adapted. Vice versa, no chicken samples were taken on 21st December – which probably explains the lack of N2s (from chicken-adapted H9N2s). Grouping "poultry swabs" together isn't reasonable, given these different adaptations. Were none of these birds present in the market on these times? I'm sympathetic to the realities of fieldwork, but I do think that these comparisons are misleading/oversimplified as currently presented in the text.

Line 392: "Given that de novo...". There's little data presented on whether the de novo assembled sequences do accurately represent those detected in poultry swabs. Whilst we see that they indeed identify the same viruses, the accuracy of the genome information below the broader clade level is not shown. For example, in Figure 3 it's unclear which are new environmental and which are new poultry swab data, and how similar these are. There are some quite long branches in some of the new data, and I would feel somewhat reassured if these weren't in the environmental samples because of the

risk of de novo assembly producing strange chimeras. It would have been useful to have some artificial pools here, e.g., comparing results from sequencing fake “environmental samples” made of pooled swabs, to those obtained from sequencing individual swabs.

Minor comments:

The title suggests that air sampling is sufficient to capture the diversity of zoonotic viruses circulating in the market. Yet there is little clarity provided that any pathogen detected here is zoonotic, with the (implied) exception of avian influenza virus. It would be useful to have some added discussion of the diversity of the zoonotic viruses observed to justify this title.

Line 24: It is wrong that environmental surveillance has not been used for zoonotic pathogens. The abstract itself defines ES as including air and cage swabs, etc. There is extensive evidence of the use of environmental sample types being used to detect zoonotic pathogens including influenza since at least the 1970s, e.g., see <https://doi.org/10.1111/tbed.13633>. This is recognised by the authors at line 77.

Line 108: Given that the markets are named, it could be useful to see the point locations on Fig S1.

Line 120: Additional details should be provided to allow reproducibility, e.g., the type of swab, type of VTM, length of vortex etc. The Direct-zol RNA Miniprep manufacturer’s protocol provides various different conditions for different sample types, and it would be useful to understand which was followed for each sample type, as I imagine that this is different for the pooled swab and the liquid samples. How the samples were pooled into libraries and how many reads per sample and sample type would also be important in the Supplementary Information. The number of PCR cycles in the Twist protocol would be helpful too.

Line 178: the R package SourceTracker will not be familiar to all readers, and it would be useful to add some brief details of the approach of this package.

Line 204: Why was no bootstrapping or other estimates of nodal support done?

Data availability: Will the consensus sequences be shared individually, at least for those from the individual hosts? Or is this just SRA? I’m glad these data will be shared, but I can’t evaluate how appropriately this has been done without access to the project or the Github.

Line 221: were any novel viruses found? Twist approaches would be able to detect novel viruses, at least for some families. What is used to define whether a virus is “found”, e.g., what if only a short contig of 200bp is present, or not all genes? This is especially important thinking about retroviruses, given the possible similarity to host DNA.

Line 231: is gull coronavirus definitely proven as respiratory, not fecal-oral?

Line 248: I don’t understand this conclusion. Figure 1C shows the number of viruses detected, but not the diversity. So it’s not clear whether more sample types would be necessary, or, e.g., if wash water contains only a subset of air sample data.

Line 277 to 280: I don’t understand why this approach was used rather than something simpler, please clarify to the reader.

Line 283 “tended to have more negative correlations”, is also unclear.

Line 306 – “performs better at detecting chicken viruses” – perhaps this conclusion could be phrased as “performs better at detecting viruses from the chickens, which were also the most abundant host in this market”, or equivalent. Whilst I note your Discussion point about this, I think it’s useful to add this note here or at least a link to the Discussion. Adding information on the total estimated relative numbers/proportion of individuals per species in each market would be critical here.

Figure 3H doesn’t appear to have any text associated with it in the legend. Should it also say N6, not H6?

Line 403: Correct to “Gamma”

Version 1:

Reviewer comments:

Reviewer #1

(Remarks to the Author)

The revisions fully address all of my concerns.

A minor note on some of the supplemental figure labels. Many of the labels from figures with corresponding tables of statistical results have this at the end:

" P values for all pairwise comparisons can be found in Supplementary File X-Y. P values are annotated as follows: P < 0.05 *; P < 0.01 **; P < 0.001 ****"

The actual table numbers are missing and the asterisks are missing from the figures.

Reviewer #3

(Remarks to the Author)

Thank you to the authors for thoroughly addressing most of my comments. The additional detail, new analyses and new figures have really improved the quality of this paper, and the ability of readers to fully understand the data being presented. I have only a few outstanding comments:

- Thanks for adding Figure S2. It is useful to see that no poultry-virus reads were identified in the negative controls. However, please add detail comment about what the negative control sample was (for example, was this a water control, processed from the point of extraction alongside all other samples analysed here? Or a completely separate run? Or

something else?). The choice of negative control has important implications for interpretation. If the negative control was not processed alongside these samples, that is a significant weakness.

(Please note that I disagree with the author's comment in the rebuttal that "sterile" air filters are unlikely to be informative as a negative control. Sterile implies inactivation of pathogens, and does not imply RNA/DNA free status. I agree that it's unlikely that filters would contain poultry pathogens, but use of a blank negative control that uses all of the plasticware and reagents and is run alongside real samples is best-practice for metagenomic studies including facilitating subsequent reanalysis of SRA data where people may not just be interested in poultry viruses. But I agree that it's not critical here - it's just critical to state what the actual choice of negative control was as noted above).

- Thanks for adding information on the number of samples per time point. Please confirm in the text that samples from individual birds each came from a different stall (i.e., the birds were randomly sampled from all birds at the market). This is important because birds from the same stall/cage are more likely to share the same viruses, compared to environmental samples from different stalls. This sampling design is valuable for the reader to understand, especially when many field studies do not do this as it's harder to sample in this way in the field.
- To be clear to the editor and the authors, my question about "were any novel viruses found" is not intended to "dismiss" viruses where only short contigs are found as the authors imply. I fully agree that short contigs can be informative. However, it is important to provide clarity to the reader about what was counted as a "detection" here and whether any effort was made to discriminate between known and novel viruses. This clarity is necessary when interpreting the manuscript for readers who may be interested in virus diversity in these settings. The rebuttal now states that they do not attempt to detect novel viruses. To aid the reader's understanding, it should be clearly stated somewhere in the manuscript - perhaps the Discussion? - that just because a virus has a close hit in nr using DIAMOND, it does not mean that the contig is 100% conclusively that virus. This is a recognised problem with metagenomic studies (I am reminded of <https://doi.org/10.1016/j.cels.2015.07.008>!).

Similarly, I would recommend removing the new text in the concluding paragraphs that states that "environmental samples enable detection of both known and novel viruses across multiple hosts and sample types".

- Thanks for stating that you now use bootstrapping on trees. However, I don't understand how removal of parental branches with bootstraps <70 was done on Figure 4 and S42/S43 – do NONE of the internal branches have bootstraps <70? I can understand how pruning of external branches and corresponding samples could have been done, but it seems unlikely that removal of internal branches with bootstraps <70 would be possible without leaving a partially unconnected tree. Please check, and make the process of this filtering clear to the reader with a caveat somewhere about the potential issues that can be caused by phylogenetic estimation on mixed length sequences where aligned regions may come from different genome sections in different viruses.
- Figure S16: the rebuttal says that "this heatmap being referenced shows abundances as ranked RPM which is not the same as raw RPM". Thanks for adding the information on rank scaling to the methods. Please can you modify the legend slightly to make this clearer to the reader? "Heatmap showing the abundance of non-poultry viruses detected, represented as reads per million" implies raw to me, even though Rank(RPM) is on the figure.
- There are a lots of typos generally in the newly added sections (e.g., "racptured" in S5 legend, "environmental" multiple times in S11). Please check.
- In the phylogenies:

o it's almost impossible to see out the tip shape in some of the phylogenies where the tip shapes are overlapping. Perhaps the trees could also be provided as separate nexus/newick formats files, to help with this?

o The tip shape legends in all trees in the main text and supplementary are hard to understand (e.g., the Figure 4 legend doesn't mention the square tip. Are the poultry samples not generated here as "new sequences"? If not new, what differentiates these "poultry" from "reference"?)

o Please make all the trees use the same tip shape for the same tip type. For example, it really confused me for a long time that in 4A, triangles are "reference" but in 4B, triangles are "new sequence". This makes it hard to read, but also inconsistent with the legend text.

o I don't really understand the red arrow or what they mean. They seem to point to quite vague points on the tree, whereas generally I think that highlighting a small clade is the intention. Consider, perhaps, changing the tip opacity for the samples from these markets instead (i.e., more transparent for samples from the reference dataset might work well, and more clearly show the clustering of different sample types from this study).

Response to reviewers' comments

Please note that the line numbers mentioned in our response refer to the revised manuscript with tracked changes (all markup). Please see our responses in blue, and the line numbering follows the clean version.

REVIEWER COMMENTS

Reviewer #1 (Remarks to the Author):

Cronin et al. examined the utility of using environmental sampling at live bird markets for surveillance of influenza and other viral pathogens of concern. Zoonotic pathogens pose a serious threat to human health, and live animal markets are especially hazardous when it comes to zoonoses. Environmental sampling is less expensive and easier logistically than directly swabbing animals. They found that most of the viruses detected from direct swabs were found through environmental sampling, although there were differences in the viruses found in each environmental sample type. Environmental sampling was better at detecting the chicken virome than the duck virome. Environmental sampling is an important surveillance tool and it is critical to understand how well it performs compared to more traditional methods

Major Issues

The usefulness and impact of this study would be greatly improved if recommendations on optimal environmental sampling were given.

We thank the reviewer for raising an important point. We agree and to address this we have now revised the manuscript to include recommendations on optimal environmental sampling. This addition has been incorporated into the Discussion section (lines 1048 to 1058). In response to this and related comments in the rebuttal, we have added a substantial number of new figures to help clarify what the optimal strategy is in this context.

In short, ES sampling should not replace but complement existing poultry swab sampling for surveillance. ES of air and cage surfaces should be prioritized, as these sample types consistently detected the highest diversity of poultry viruses (Figure 1). However, other ES types should also be included. As demonstrated in Supplementary Figure 11, every different environmental sample type is implicated at least once in capturing a poultry virus no other environmental sample type could detect (see next comment). Figure 2 also highlights that ES performs better in recapturing chicken viruses than duck viruses. This analysis suggest that in addition to ES, limited but targeted oropharyngeal and cloacal swabs especially from ducks remain important for detecting species-specific viruses that may not persist in the environment. While wash water detected the lowest number of poultry viruses (species richness) compared to other sample types it was the shortest distance (Jaccard beta-diversity) from oropharyngeal and cloacal swabs in ducks (Figure 2). This indicates that while wash water is less diverse it shared a higher proportion of their fewer viruses with duck samples highlighting its importance. The same point can be made about drinking water which tended to detect less poultry viruses than other ES (Figure 1). However samples in this group were the shortest distance from the chicken oropharyngeal median centroid again suggesting that it shared a higher proportion of their fewer viruses with chicken oropharyngeal swabs. While there are multiple layers to all of this we believe it is clear that a hybrid approach is most beneficial to broaden surveillance coverage. Multiple ES types should be used in combination with poultry swabs (ducks prioritised).

This optimal sampling strategy described here is of course context dependent. Other human animal interfaces have different animal species, different viruses and ES needs to be validated/evaluated in other settings as the optimal strategy may be different.

In lines 247-250 it states that "a variety of sample types are required to accurately capture the most viral viruses from poultry". It would be helpful if this was followed by how combining specific environmental sampling types increases the ability to recapture the virome from the direct swabs.

We would like to thank the reviewer for their constructive feedback. As briefly mentioned in the previous comment we have conducted a new analysis presented in Supplementary Figure 11 to address this. These figures display the total number of viruses detected in each poultry swab across timepoints, alongside a stacked bar plot representing detections in environmental samples. If a virus was identified in multiple environmental sample types at a given timepoint, it is shown in the lower portion of the bar labelled in the legend as >2 ES (coloured in grey). In contrast, viruses detected exclusively in a single environmental sample type at a given timepoint are represented in the upper portion of the bar and colour coded accordingly. Every different environmental sample type is implicated at least once in capturing a poultry virus no other sample type could detect. Consistent with our original statement, this analysis demonstrates that while the majority of viruses can be detected across multiple environmental sample types, a subset of viruses are uniquely identified in specific samples. Therefore, incorporating multiple sample types enhances the overall recovery of viral diversity.

In addition, when considering the total number of poultry viruses detected (not just those recaptured) we show in Figure 1B that air from the slaughter and holding areas, as well as cage swabs have a higher alpha-diversity than other groups. In Figure 2D we show that these three ES types are the shortest distance from the median centroid of chicken cloacal swabs. While in Figure 2C cage swabs are shown as the group with the second shortest distance from oropharyngeal swabs in chickens. Together these findings based on both alpha-diversity and beta-diversity analysis indicate that air (slaughter and holding area) and cage swabs enhance poultry virus surveillance and thus must be included. In Figure 2E-F we see that wash water was the shortest distance from oropharyngeal and cloacal swabs in ducks suggesting it performed best at detecting viruses from this host. This is important as in Figure 1 and associated Supplementary we often see wash water detecting less viruses than other ES types. This tells a story of quality over quantity while wash water is less diverse (alpha-diversity) it shared a higher proportion of their fewer viruses with duck samples. This analysis also captures temporal consistency, as it incorporates data from all twelve sampling visits to the live bird market. The same can be said for drinking water which was the shortest distance from oropharyngeal swabs in chickens (Figure 2C).

Five out of six ES types in this study contributed meaningfully to poultry virus detection, supporting the inclusion of diverse sample types to maximize surveillance performance. Based on these findings and comments from other reviewers we have now described the optimal sampling strategy in the discussion on lines 1048 to 1058.

We believe the new figures and analysis better supports the statement. The manuscript text has also been updated at lines 438-445, 458-463 and 510-552 describing these findings.

The title doesn't accurately reflect the conclusions. Air sampling alone isn't sufficient to detect the range of viruses and environmental sampling better reflects chicken than duck viromes. Something like "Environmental sampling closely recapitulates direct chicken swabs for avian viruses circulating in live poultry markets" would be better.

We agree with the reviewer and have now updated the title accordingly. We have changed air to environmental sampling. We have however not opted to specifically mention chickens rather than ducks. Yes the performance for chickens was better than ducks but as mentioned in response to the first comment this approach is meant to "enhance" not replace. With respect to this we believe the new title accurately reflects the content of the manuscript.

The github link doesn't currently work.

This has now been resolved. The GitHub is publicly available with all relevant data and R scripts. Please note that the link has been updated to - <https://github.com/PeterCroninX/air-sampling-zoonotic-virus>.

The figures overall are very nice, but there are a few issues that impede the ability to interpret them. All of the air samples are colored the same despite being treated as separate categories for analysis. This

is an issue when not all data points are labeled, such as in Figure 1C, Supplementary Figure 4, and Supplementary Figure 7.

We greatly appreciate the reviewer's detailed comments. In response, we have updated all figures in the manuscript with each of the 10 groups having a distinct colour.

The following issues are more minor.

Within the figures the order of the samples isn't consistent and takes a little extra effort to read (i.e. Figure 2 D-G and Figure 3H). For Figure 2A, the circle is unnecessary, and the figure would be easier to interpret if it was linear. Figure 4 – It would be easier to see what is going on if the tip shapes were a bit smaller or slightly transparent.

We thank the reviewer for their helpful suggestions. We understand the concern regarding the ordering of groups in Figures 2D–G and 3H. The figures being referred to are now Figure 2C-F in the latest draft. The ordering of these figures was intentional and we believe it is important to maintain this order for good reason. This is because the groups are ranked by their distance to the median centroid, with the group closest to the centroid appearing first. Given the median centroid is calculated solely based off poultry swabs this group always appears first. The second group in each case corresponds to the environmental sample that is the shortest distance from the median centroid of the respective poultry swab (in other words most similar based on the PCoA of beta-diversity). We acknowledge that this rationale was not clearly explained in the original manuscript, and we have now updated the figure legend for Figure 2 (lines 1472-1473) to clarify this. We believe having a standardised order in this case only causes confusion as it may not be clear which group is most closely associated with each poultry swab type. However, outside of this one case we have ensured that the ordering of all other figures is consistent throughout the manuscript.

For Figure 4C (previously Figure 3H) we have now also updated the order. From top to bottom the heatmap is broken down by date while poultry swabs always appear first followed by environmental. Furthermore, circular heatmaps have been replaced with more conventional linear representations, and the phylogenetic trees in Figures 4 have been revised by adjusting the tip shapes (shapes made smaller and also distinct shape for poultry and environment) for improved clarity. We agree with the reviewer that these modifications enhance the overall readability of the figures.

Figure 4 being referred to here has been removed in line with comments from other reviewers and new analysis included this time around (associated with Figure 3). However, phylogenetic trees are still included for avian influenza (now in Figure 4). With respect to this comment we have now changed the aesthetics of the trees reducing tip size.

Minor issues

Line 298 -301: Exactly how many additional avian viruses were detected. Supplementary Figure 5 is labelled as Supplementary Figure 4.

We thank the reviewer for their feedback. Specifically, 5 additional poultry viruses were detected, and we have updated the manuscript accordingly at lines 475 to clarify this.

We would like to emphasize what we originally meant by "additional avian viruses" in this context, especially as we have added several new figures and made substantial revisions to the original draft. In the original draft, "additional avian viruses" refers to viruses detected in environmental samples that were never detected in poultry swabs at any timepoint. We identified 5 "additional" poultry viruses by this definition. This figure is still included as Supplementary Figure 16 and associated text shown at lines 471 to 481.

In the latest draft of the manuscript when we refer to "poultry viruses," we specifically mean the 84 viruses that were detected in at least one poultry swab at one or more timepoints in this study. Except for Supplementary Figure 16 all analysis presented in the study is based on these viruses. Importantly, there are many instances where ES detects some of these viruses but poultry swabs fail to do so. It's important to mention this as in the original draft much of the analysis focused on the ability of ES to

directly recapture exactly what was present in poultry swabs with little attention given to this detail. In line with other reviewers comments we have updated all figures to reflect this.

We have also ensured that all Supplementary Figures are correctly labelled.

Reviewer #2 (Remarks to the Author):

The authors present a new, robust and more inclusive approach to the surveillance of avian viral pathogens in high-risk settings, environmental metagenomic surveillance, which provides a better overview of the viral diversity in such settings due to its ability to detect several pathogens at the same time, using only minimal resources in the collection of environmental samples at the same time.

The authors demonstrate that the same pathogens can be detected accurately by metagenomics on environmental samples as by metagenomics on poultry samples. However, the authors also acknowledge that several different environmental sample types may need to be combined to achieve a similar level of coverage to poultry samples.

The introduction covers all the important aspects related to different surveillance strategies, but it could provide some more context about the specific study location. The proposed approach is methodologically and technically coherent and demonstrates the authors' expertise in the application of the described laboratory and (bio-)statistical methods. All methods are described in detail and reproducible. The results are underpinned by numerous useful illustrations. The discussion addresses all the key points raised in the text, although some limitations in the interpretation and follow-up of the results need to be discussed. For example, it remains unclear which specific intervention strategies can be chosen based on the results and where they should be applied. In addition, aim to prove the superiority of the proposed approach over "conventional pathogen surveillance", but it remains unclear what this baseline is.

Introduction:

-Could you provide a short summary of studies looking into avian viral pathogens in the study location by "conventional pathogen surveillance"?

-Could you please explain what your understanding of "conventional pathogen surveillance", to which you are comparing your approach, is? At the beginning of the discussion, you seem to compare your approach to "targeted probe-enriched metagenomics on direct poultry swab samples", but it does not become clear from the text before.

We thank the reviewer for their input. We agree this was not clear in the original draft. Firstly we have updated the introduction at lines 113 to 115 citing studies which have used convention pathogen surveillance methods in Cambodia.

When we refer to conventional or traditional pathogen surveillance in the manuscript we were referring to the random collection of poultry swabs (both oropharyngeal and cloacal) which are then typically subject to PCR-based methods for the screening of one or few pathogens (e.g., Influenza A and Coronaviruses). While this approach offers high sensitivity and specificity it is limited in the number of viruses it can detect. Metagenomics has emerged as a method with a much broader target range for virus identification and surveillance, allowing detection of a diverse array of known and potentially novel pathogens. Independent of the surveillance method (PCR/Metagenomics) the conventional sampling strategy has always been random poultry swab sampling which may miss circulating pathogens if the selected bird is not infected when sampled, thus limiting the probability of detection. All of this is mentioned in the introduction at lines 111 to 126. With respect to all of this, the current study aimed to compare targeted virus capture probe-based metagenomics on several different ES types versus the conventional sampling strategy of random poultry swabs collected from LBMs.

We agree that in the original draft it was unclear what we were comparing against. We have now updated the manuscript at lines 36 to 37 in the abstract and lines 127 to 129 in the introduction to specifically state what we are comparing against. We believe these changes now make it clear to the reader what our approach is.

Methodology:

-Why were exactly those two live bird markets chosen?

-In some instances throat swabs are mentioned, in others oropharyngeal swabs. Which ones were used and why? Oropharyngeal swabs would be the preferred choice for the detection of avian influenza viruses. Was viral transport medium used and what were the transport conditions on the way to the laboratory? All this may impact the probability of extracting sufficient high-quality RNA for sequencing.

We thank the reviewer for their feedback. Orussey market, located in Phnom Penh, is the largest live bird market in Cambodia and serves as a major hub for poultry trade. Birds are brought to the market from a broad geographic area, leading to high animal density and extensive inter-regional mixing. This centralization of poultry trade increases the potential for pathogen transmission and amplification which makes Orussey market a key site for avian influenza virus surveillance and research in Cambodia. In contrast, Takeo market was selected as a study site due to its smaller scale, lower trade volume, and primarily regional bird sourcing including cross border poultry trade. It is representative of the many small-scale, decentralized live bird markets dispersed throughout Cambodia. Studying such a market provides critical insight into local transmission dynamics and complements findings from larger, urban centres like Phnom Penh. Together, these sites offer a broader understanding of virus circulation across different market types. We have now updated the methods section at lines 138 to 147 highlighting the rationale for surveillance in both of these markets.

Originally when we used the word "throat" we were referring to oropharyngeal swabs. We have now removed the word "throat" from the manuscript replacing with oropharyngeal.

We have also updated the methods at line 166 to 178 describing the sample transport medium/conditions used in this study. Specifically, the VTM used in this study was prepared in-house and consists of 2.95% tryptose phosphate broth, 145 mM NaCl, 5% gelatin, 54 mM Amphotericin B, 106 U/L penicillin-streptomycin, and 80 mg/L gentamicin. All samples were transported immediately and on ice to the lab located in Phnom Penh. No liquid nitrogen was used to avoid a freeze-thaw cycle.

Discussion:

-It would be useful to discuss how the use of different sample collection methods, sample types and transport conditions may have impacted amount and quality of the RNA extracted.

Thank you for this valuable suggestion. We agree that variations in sample collection methods, sample types, and transport conditions can significantly impact the amount and integrity of RNA recovered, particularly when comparing environmental samples to direct poultry swabs.

To address this, we have revised the Discussion section (lines 1000–1004) to acknowledge these limitations and their potential influence on downstream sequencing and virus detection outcomes. Specifically, we now note that environmental samples, such as air, water, and surface swabs, may yield lower viral loads and be more susceptible to RNA degradation or inhibitory substances, which could partially account for their lower virus detection rates relative to direct avian samples. These differences highlight both the technical challenges and interpretive considerations when integrating environmental metagenomics with host-based surveillance.

-Why in your opinion are air samples performing better than other environmental samples?

We believe that the relatively strong performance of air samples may be attributed to their low background contamination. Compared to other environmental samples (e.g., animal swabs or feces), air typically contains less bacterial and host material, reducing nonspecific nucleic acids and enhancing viral enrichment and sequencing efficiency. We have now updated the manuscript in the discussion at lines 958 to 961 stating why we believe air performed better than other sample types.

-Why do you think is the approach better at detecting chicken than duck viruses?

One likely explanation is that a significantly higher number of chickens pass through these markets on a daily basis compared to ducks, which reflects the greater demand for chicken meat relative to duck

meat. As a result, the market environment is more heavily contaminated with chicken-associated viruses, leading to increased detection rates. The lower number of ducks means there is a lower chance of the environment becoming saturated with viruses from this host. We have addressed this in the discussion at lines 1018-1032.

-Why do you think carcass wash water provided statistically significantly different results?

Wash water likely captures oropharyngeal viruses efficiently due to the release of respiratory fluids during slaughter, direct rinsing of the upper airway, and the pooling effect across multiple carcasses, resulting in a concentrated representation of respiratory virus in this sample type. We have now mentioned this in the discussion at lines 972 to 1000.

-The authors claim that the proposed approach is more cost-effective. Is there any evidence or an example of what kind of resources may be saved?

We appreciate this insightful question. To clarify and support our claim, we have expanded the Discussion section (lines 1094–1112) to explain how environmental sampling, particularly when paired with metagenomics, can substantially reduce both logistical and financial costs associated with traditional surveillance. Rather than individually sampling and testing potentially thousands of birds to characterize viral circulation in live-animal markets, a small number of environmental samples, such as air, water, or surface swabs, can capture viral signals from multiple species and sources simultaneously. This pooled exposure effect reduces the number of samples that must be collected, transported, processed, and sequenced, resulting in significant savings in labour, consumables, and laboratory time. Furthermore, the use of the CZ ID platform, which is open-source and designed to function with minimal computational infrastructure and bioinformatics expertise, eliminates the need for specialized software licenses or high-performance computing resources, further lowering barriers to implementation. Taken together, this integrated approach enables faster, broader, and more cost-effective surveillance, particularly in resource-limited settings such as Southeast Asia. These revisions have now been reflected in the updated manuscript.

-Can the proposed approach realistically be implemented on a wider scale in Cambodia or similar settings? Do laboratories in such settings typically have the necessary equipment and expertise to carry out this type of analysis compared to, e.g., conventional PCR testing?

We thank the reviewer for raising this important point. We believe, and have shown, the proposed approach is both realistic and feasible for broader implementation in Cambodia and similar resource-limited settings across Southeast Asia. All wet lab procedures and sequencing in this study were conducted at the Institut Pasteur du Cambodge (IPC), a World Health Organization (WHO) Reference Laboratory equipped with an Illumina MiSeq platform. This demonstrates that sequencing-based surveillance can be successfully conducted within-country, without requiring outsourcing to international facilities.

However, even in settings where in-country sequencing capacity is not yet established, next-generation sequencing is increasingly accessible through external commercial or institutional services. These platforms offer relatively rapid turnaround times and continually decreasing costs, in part due to the long-surpassed limits of Moore's Law in sequencing technology. The cost per genome and data processing time have dropped significantly over the past decade, making external NGS services a viable and scalable option for many low- and middle-income countries.

To further lower the technical barrier, we used the CZ ID platform for downstream analysis. CZ ID is open-source, freely available, and optimized to run on minimal computational infrastructure with limited bioinformatics expertise—making the entire workflow accessible to labs without dedicated bioinformatics personnel or high-performance computing. While conventional PCR testing remains valuable for targeted diagnostics, metagenomic sequencing offers a broader, unbiased detection capability that can identify both known and novel pathogens. These additions have been incorporated into the revised Discussion section (lines 1106–1112).

-How can the proposed approach complement rather than replace existing approaches?

We believe the proposed approach is best positioned as a complement rather than a replacement for existing surveillance methods. Our findings show that environmental samples tend to have higher alpha diversity for poultry viruses compared to poultry swabs, allowing for the detection of a broader range of viral species. However, environmental samples were only able to recapture approximately 70–90% of the viruses detected in poultry swabs. This indicates that while environmental metagenomics offers broad coverage, it may miss certain viruses present in individual hosts. In addition, we found that environmental metagenomics performed better for chickens than for ducks, likely due to the greater number of chickens present at the market. This suggests that direct sampling may be particularly important for animal groups present in lower numbers at human-animal interfaces to ensure their viral diversity is adequately captured. Therefore, integrating environmental sampling with poultry swab metagenomics provides a more comprehensive surveillance strategy casting a wider net and maximizing the likelihood of detecting diverse pathogens circulating in live poultry markets. We have updated the manuscript in the discussion section at lines 1057 to 1066 to address this.

-The authors claim that the proposed approach allows for more rapid and effective risk mitigation. Which specific mitigation strategies can be chosen based on the results and where should they be applied? It would be important to discuss here that the proposed approach is not able to identify the infected population within the setting, which would be necessary to design targeted mitigation strategies, e.g. for specific poultry production systems. Given that infection cannot be traced back to a certain population, intervention at the source is not possible. For example, avian influenza viruses are found in most live bird markets in Southeast Asia, so how would a positive test result be used?

We thank the reviewer for this insightful and important comment. We agree that environmental metagenomic surveillance does not allow for identification of specific infected animals or tracing infections to particular production systems. However, we believe the approach still enables timely and actionable mitigation strategies in high-risk settings such as live bird markets, especially when applied in combination with targeted follow-up actions.

To clarify this point, we have added a paragraph to the Discussion (lines 1067–1093) outlining specific, realistic interventions that can be guided by environmental data. For instance, detection of elevated viral loads on cage swabs may prompt enhanced disinfection of cages and holding areas. High levels of viral contamination in air samples can support operational changes such as covering defeathering machines or modifying ventilation systems. Repeat environmental sampling can then be used to assess the effectiveness of these interventions and determine whether further measures, such as poultry quarantine (e.g., 48–72 hours) or temporary market closure, may be warranted.

Furthermore, when novel or high-consequence viruses are detected, targeted responses such as confirmatory PCR, host-specific swabbing, or upstream supply chain tracing can be initiated. In this context, environmental surveillance serves as a scalable and cost-effective early warning system. While it cannot pinpoint the infection source at the individual or flock level, it enables risk-based decision-making and complements existing animal surveillance strategies. These clarifications have now been incorporated into the revised manuscript.

Other comments:

-Is the title appropriate given that the authors are talking about more environmental samples than just air samples?

We agree and we have now updated the title to more accurately reflect the findings of the study.

-The data, code and analysis files under <https://github.com/PeterCx/air-sampling-zoonotic-virus> are not available.

This has now been updated accordingly. The GitHub is publicly available with all relevant data and R scripts. This link has been updated to - <https://github.com/PeterCroninX/air-sampling-zoonotic-virus>.

-Table 1 is not mentioned in the text.

Table 1 has also been updated to a more detailed version included as Supplementary Table 1 which shows the number of samples collected at each timepoint and market. It is now referred to in the text at lines 157, 162 and 166.

-Lines 48-49: please check for some newer references on wastewater surveillance.

We have now added several new references at line 70 for wastewater surveillance as requested.

-Line 82: the results do not indicate at which point along the poultry supply chain interventions are appropriate.

This has now been addressed in a previous response above (regarding risk mitigation). We have included details of this in the discussion at lines 1067–1093.

-Lines 292-294: does this include pathogens not typically associated with the respective host (chickens or ducks)?

This analysis was conducted for all 84 viruses detected on at least one market visit and in at least one poultry swab across the study. The viruses in question can be seen in Figure 1A and mentioned in the text at lines 364-365. Some of these viruses were only found once and in a lower abundance so they may not be typically associated with the host.

-Lines 341-344: maybe the season played a role? Or some sort of festivity?

We thank the reviewer for raising an important point. It has been previously shown that there is an increase in avian influenza virus at live-bird markets in Cambodia which directly corresponds to the four major Cambodia festival periods. However when these samples were taken (December 2022) this would have been a month after the November Bon Om Touk (water festival) and two months before February Lunar New Year. Thus there is no evidence it is associated with a festivity.

We have added new text to the manuscript at lines 777 to 779.

Reviewer #3 (Remarks to the Author):

Overall comments:

This paper investigates approaches for virus surveillance in live bird markets. It describes the comparison of viruses detected in cloacal and throat swabs from poultry to those detected from environmental samples (air, carcass wash, cage swabs). Similar viruses were found in the environmental samples compared to the poultry swabs, including some viruses that were likely not from the targeted poultry species (chickens and ducks). This indicates that environmental surveillance approaches could be useful for conducting surveillance of viruses in live animal market settings.

Whilst use of environmental surveillance for zoonotic and veterinary pathogens is not new, this study provides a potentially powerful direct comparison of results from environmental samples and swabs. There are however currently some major weaknesses in the study design or reporting. I detail these concerns below, but key areas include a severe lack of clarity and detail in reporting of methods (e.g., numbers of samples per time point), apparent absence of negative control samples, and phylogenetic estimation from virus contigs that have been created from environmental samples that likely represent chimeras of virus genomes from many different individuals. I am also concerned about some elements of statistical analysis, such as little effort to statistically control for differences in the number and type of samples per time point. I hope that the authors can address these concerns.

Major comments:

What appropriate blank negative controls were sequenced here (e.g., a blank swab taken in the market, an unused air filter)? Given that the aim is to compare different sample types, it would be extremely important here to confirm that the diversity of viruses observed are not the result of contamination between samples during sequencing. Metaviromic studies can also easily detect viruses present via other contamination in the lab or reagents, and this can be excluded with appropriate controls.

We appreciate the reviewer's concern regarding potential contamination and the importance of including appropriate negative controls in virus metagenomic studies. Originally we sequenced five negative control samples to monitor for potential contamination. These controls were not included in the original manuscript, as no evidence of cross-contamination or background viral signal was detected. In line with other virus metagenomic studies we only counted reads if they mapped to an assembled contig (methods section lines 240 to 241). Of the five negative controls sequenced, only one produced assembled contigs, which were all identified as *Totiviridae_sp.* a virus known to infect fungi. *Saccharomyces cerevisiae* is a known host of this virus species. This virus was not detected in any other sample in our dataset. Ultimately, this means that aside from a single fungal virus, all negative controls showed zero counts for every species detected whether environmental or poultry-associated. This rules out any contamination from the lab/reagents. To address the reviewers concerns we have included a new Supplementary Figure 2 in the manuscript which is a heatmap showing all species detected in the study. We have also updated the manuscript at lines 211 to 217 in the describe these findings.

However, the specific examples suggested (i.e., a "blank swab taken in the market" or an "unused air filter") are difficult to interpret in the context of our study. Once a swab is exposed to the market environment, it can no longer be considered a true negative control, and therefore cannot provide a meaningful baseline. Similarly, the suggestion of sequencing an unused air filter is unlikely to be informative, as such filter commercially manufactured by Thermo Fisher Scientific are supposedly sterile. Even if these filters are contaminated with viral nucleic acid during the manufacturing process it is highly unlikely they are contaminated specifically with a plethora of poultry pathogens. In our view, such materials do not constitute appropriate or realistic negative controls for this type of work.

Line 111: The number of samples collected is not present in the methods. The information is roughly summarised in Table 1, but this is never referred to in the text. In any case, it surely needs to be more detailed, i.e., broken down per market, per time point, and per sample type and species. It's not clear how bird samples/stalls were selected – were these taken from both the slaughter and the poultry holding area as well? How many stalls? Without this information, it's impossible to evaluate whether environmental sampling is better, or whether it simply targets broader or different locations.

We thank the reviewer for their valuable feedback. We have now clarified the sampling strategy in the methods section at lines 147 to 166. Table 1 has been moved to Supplementary Table 1 and shows the number of samples collected for each group at each LBM visit. Birds and cages were chosen randomly for sample collection. It's important to note that poultry swabs are only taken from live birds. Typically at the market once birds arrive they are kept in the holding area either in cages or occasionally on the ground adjacent cages. Once live birds are transferred to the slaughter area they are culled almost immediately. With respect to this, poultry swabs can only be obtained from live birds within the holding area. Furthermore, like poultry swabs, cage swabs in this study were pooled. 5 cages were randomly selected and swabbed at each LBM visit. Five drinking water samples were also randomly collected. Drinking water is contained within the cage but not all cages typically have drinking water. With respect to this drinking water was often not collected from the same cages that were swabbed. Up to two carcass wash water sample was collected at each timepoint because typically there is only no more than two carcass wash water containers in the slaughter area of LBM.

Line 157: The bioinformatic information is not sufficiently detailed. Please provide information on the mapping approaches (e.g., what is the reference database used here), as well as information on sequence length filtering etc (e.g., what parameters). What number of reads or contig length is considered a "detection", and how does this compare to negative controls?

We thank the reviewer for their response. We agree with the reviewer and have now made several changes to the methods section providing greater detail for the bioinformatics methods used at lines 220 and 246. We have also included Supplementary Table 2 which shows the number of reads retained after each QC step in the CZ ID pipeline. The reason that the bioinformatic methods were not sufficiently detailed in the original manuscript was because we used the CZ ID pipeline. This is an open source free to use platform which has previously been published. In the original manuscript we provided references and a link to this pipeline. The full details are explained here (<https://chanzuckerberg.zendesk.com/hc/en-us/articles/360034790554-Illumina-Pipeline-Details>). LBMs in Southeast Asia are often located in areas where access to substantial computational resources for data analysis is limited. This is a key reason why we selected this pipeline, as the framework is straightforward to apply and adaptable to resource-limited settings.

Briefly, raw sequencing reads were first processed with fastp to remove adapter sequences and filtered based on quality and a minimum read length threshold of 35 bp. Host-derived sequences were then excluded by aligning quality-controlled reads to the human, chicken, and duck reference genomes using Bowtie2. The remaining non-host reads were assembled *de novo* into contigs using SPAdes. Taxonomic classification of contigs was performed via DIAMOND alignment against the NCBI Non-Redundant Protein database. Finally, reads were mapped back to the assembled contigs using Bowtie2 for abundance estimation.

For the majority of analysis in this manuscript contigs of any length were included. However, raw abundance estimations from CZ ID were subject to a 0.1% relative abundance filter in order to remove false positives and more robustly capture the proportion of viral signal within each sample. Thus a detection in this study includes contigs of any length but is only included if greater than 0.1% of reads map to this contig in at least one sample. The only exception to this was for phylogenetic tree construction where contigs had to be 200bp or greater to be included. This was done to avoid biasing phylogenetic inference as short fragments may lack sufficient phylogenetic signal and can distort tree topology. We acknowledge a mistake in the original draft of this manuscript as we had written in the results section that contigs of greater than 200bp were used for mapping poultry contigs to the environment. However the 200bp cutoff was only used for phylogenetic trees. Poultry contigs were mapped to ES contigs if they passed the filtering criteria mentioned above.

With respect to all of this information, numerous changes have been made to the methods section at lines 220 to 339.

Negative controls have now been addressed in response to the first comment.

Line 192: GISAID is mentioned and sequences are used in phylogenies. Yet there is no acknowledgements table of the submitting and originating labs, or other equivalent evidence of information to appropriately acknowledge the authors sharing data on this platform. This would be required according to the terms of use of GISAID. (To be clear, I recognise that NCBI data does not require such acknowledgement according to the terms of use).

We have now included GISAID acknowledgments for all sequences used in Supplementary Table 5.

Paragraphs line 152 and line 193: It is problematic to build contigs from environmental or pooled data and rely on this for phylogenetic tree estimation. This is because chimeras that represent similar infections in many different hosts can be easily generated. At a minimum, it would be needed to perform bootstraps and provide support for grouping of environmental and non-environmental samples, and provide some comment on the limitation of this approach in terms of possible inaccuracies in estimated divergence and the potential challenges of relying on chimera sequences for public and veterinary health surveillance.

We thank the reviewer for raising this important point. We are aware of this limitation which is incredibly difficult to resolve. Originally the phylogenetic tree estimation was presented as a method of showing similarity between poultry and ES contigs for important viruses while also allowing us to visualise what subtype or clade of each virus that was in circulation at the market. Based on this and other comments we have conducted a new analysis presented in Figure 3 and associated Supplementary Figures 17-

41 which is an alternative method of showing similarity between contigs. We believe this revised analysis is more comprehensive and clearly demonstrates strong concordance between poultry and environmental contigs in terms of gene presence, gene coverage, and sequence identity.

We first identified the top DIAMOND blastx hit for each poultry contig after taxonomic assignment. Corresponding reference genes were retrieved from GenBank to build a custom database of approximately 4,000 poultry virus genes. A full list of GenBank accessions and associated annotations (genus, species, and gene type) is provided in Supplementary Table 3. All contigs that passed filtering (as previously described) were then mapped to this reference gene set to calculate coverage and percent identity as reported by DIAMOND (Supplementary Table 4). Only DIAMOND hits with a significant e-value threshold greater than $1e-5$ were retained for downstream analysis. Where poultry and ES contigs had significant hits against the same Genbank entry the difference in sequence identity and coverage was reported. If no contig in a given sample aligned to a gene with an e-value $\leq 1e-5$, the gene was considered undetected in that sample. Ultimately this new analysis shows that environmental contigs align to the majority of poultry virus genes with little difference in coverage or percent identity. This analysis is described in the new results section from lines 612 to 753.

This gene-level mapping approach provides a clearer and more conservative measure of genetic similarity than phylogenetic trees. However, we have opted to retain phylogenetic trees only for influenza A and these trees were rebuilt with 1000 bootstrap replicates, and we display only branches whose parent node is supported by $\geq 70\%$ bootstrap, ensuring clade assignments are more robust. The influenza trees therefore serve solely to contextualise circulating clades at the LBMs, while the gene-level analysis delivers the principal evidence of sequence concordance. The methods have been updated accordingly at lines 336 to 338.

While this analysis does not completely resolve the possibility of chimeras from closely related strains we have included this as a limitation in the discussion from lines 1038 to 1044. Where the detected sequence divergence of key viruses could have pathogenic implications, targeted strain-specific PCR assays should be undertaken to independently verify the assembly-derived variants

Line 193: It's unclear whether the authors used alignments that included both full length CDS and also shorter sequences (200bp) when building phylogenies. This could deeply skew phylogenetic tree estimation and phylogenetic placement. It is possible that better handling of these data is implied by "manually curated and.... trimmed to ensure alignment accuracy". Please provide additional details of this.

We thank the reviewer for this important observation. We confirm that all contigs ≥ 200 bp were included in phylogenetic analyses. Our intention was not to perform full-length phylogenetic reconstruction, but rather to illustrate that viral sequences identified across sample types are separated by short phylogenetic distances. Thus showing that environmental contigs closely resemble those from poultry swabs. While we do have a number of full-length genes assembled, we opted not to exclude shorter contigs, as we believed they still contribute meaningful data to the overall signal.

Importantly, IQ-TREE constructs trees based on a per-column maximum likelihood approach, and is specifically designed to accommodate partial sequences without systematically biasing topology, provided alignments are well curated. The presence of shorter sequences may affect resolution at deeper nodes but does not deeply skew tree estimation when the focus is on relative placement among closely related sequences.

We have however reconstructed the influenza phylogenetic trees this using 1000 bootstrap replicates and only parent branches with support $>70\%$ were included in the final tree. This is now mentioned in methods at line 336 to 338.

Regarding alignment quality, our previous phrasing ("manually curated and trimmed to ensure alignment accuracy") referred to the removal of highly divergent regions suspected to result from sequencing artifacts. We used a more reproducible process using the tool trimAl (<https://github.com/inab/trimal>), which removes poorly aligned or divergent positions based on objective criteria. Following this, we manually reviewed the alignments to confirm overall quality and to ensure

no major misalignments remained. The methods has been updated accordingly at lines 321 to 334 to provide further details of this.

Ethics: How were human reads screened in and removed from the metagenomic data, particularly the air sampling data, before submission to GenBank? Bait capture would not necessarily exclude these.

We thank the reviewer for their comments. The TWIST CVRP used in this study uses probes designed to target viruses only so host nucleic acid should be low. Nonetheless, the CZ ID pipeline removes host reads specific to the selected specimen but also automatically filters out human-derived sequences. This detail has now been incorporated into the methods section at lines 225 to 227.

Figure 1: it's a shame that Figure 1A isn't broken down into different sample types. The difference between a carcass wash – which will presumably also pick up blood-borne pathogens – and an air sampler is quite profound. I'm not convinced that directly comparing throat/cloacal samples to the whole collection of these other sample types is very fair here, especially given the number of samples per group isn't known.

We thank the reviewer for their comment. In response, we have now included new Supplementary Figures 5–8, which presents the same data shown in Figure 1A but broken down by individual sample type as requested. This provides a clearer view of virus detection patterns with respect to the different poultry swab types. Importantly, we have updated the heatmap presentation to address a limitation in the original version, which focused only on the ability of environmental samples to recapture viruses detected in poultry swabs. The revised figures now also highlight viruses that were uniquely detected in environmental samples, offering a more comprehensive view of the dataset.

While we agree that this breakdown provides useful granularity, we also recognize that it adds substantial complexity given the large number of groups and timepoints. For this reason, Figure 1A in the main text was originally designed to address a focused question whether poultry viruses can be detected in environmental samples by providing a simplified summary. We continue to believe that retaining this summary in the main figure is helpful for clarity, especially as more detailed breakdowns are now available in the supplementary material. In light of the new Supplementary Figures and updating of Figure 1A we have made a substantial number of text changes at lines 362 to 375.

We have also broken down Figure 1C (now Supplementary Figure 4 in the latest draft) by poultry virus so each comparison can be better visualised.

Figure S1 – some of these viruses have only 1 RPM. How many million reads were generated per sample? There must be some minimal cut-off of RPM below which these data cannot be trusted. What do the sequencing negative controls suggest? Avian viruses could presumably be from other birds, e.g., wild birds or domestic geese.

We thank the reviewer for their comments. We presume the reviewer is referring to Supplementary Figure 7 (not 1?). Nonetheless this is now presented as Supplementary Figure 16 in the latest draft. We agree that the approach used in this figure was not described originally we have made changes to address it.

Firstly this heatmap being referenced shows abundances as ranked RPM which is not the same as raw RPM. For a given array of values 'x' of length k, denoted by $x = \{x_1, x_2, x_3, x_4, \dots, x_k\}$, the values were first ranked. The corresponding rank-scaled values, denoted by $x^R = \{x_1^R, x_2^R, x_3^R, x_4^R, \dots, x_k^R\}$, were then calculated using the formula: $x_i^R = [\text{Rank}(x_i) - \text{Minimum}(\text{Rank}(x))] / [\text{Maximum}(\text{Rank}(x)) - \text{Minimum}(\text{Rank}(x))]$. This transformation involves ranking the original values and rescaling those ranks to fall within the [0, 1] range. First, the input vector x is replaced by its ranks, where the smallest value gets rank 1, the next gets rank 2, and so on; ties are assigned the average of their ranks. These ranks are then normalized using min-max scaling so that the lowest rank becomes 0, the highest becomes 1, and all other values are proportionally distributed in between. The function returns this vector of scaled rank values, allowing for comparison of relative positions across a consistent 0–1 scale. It is simply a method to normalise data for visualisation on a heatmap. In the case of Supplementary Figure 7 this

rank scaling was done only on the non-poultry viruses shown in figure. In this case many of the values appear as rank (RPM) = 1 because they were the only non-poultry virus detected across all environmental samples. Ultimately, this approach serves as a way to scale the data for heatmap visualization. Using raw RPM values would be impractical, as one virus may appear at 50 RPM while another exceeds 250,000 RPM. Without applying a scaling method, it would be impossible to represent this range meaningfully using a single colour palette. This information about rank scaling has been added to the methods section at lines 284 to 290.

It's important to mention that the smallest RPM value included in this study after filtering for false positives (described below) is over 500 RPM. With respect to the total read counts we have now added a Supplementary Table 2 which highlights to the reader the total raw reads and total reads after QC (including host/human read removal).

We fully agree that low-abundance viral detections require careful interpretation and that applying a minimum threshold is essential to avoid false positives. To address this, we applied a more stringent 0.1% relative abundance filter, which more robustly captures the proportion of viral signal within each sample. This method has been described in relation to several comments. We have now updated the methods section at lines 243 to 246 to explain this.

Negative controls have now been addressed in response to the first comment.

Yes poultry viruses detected in Supplementary Figure 7 could presumably be from other birds. This highlights how environmental sampling is robust and can detect viruses which would otherwise be missed by poultry sampling. This is surely a positive finding especially given that we have shown there to be no contamination.

Discussion: I was surprised that the discussion didn't mention many limitations or weaknesses, especially compared to existing sampling approaches. A key limitation would surely be the inability to quantify prevalence per market. In addition, there would be an inability to detect reassortment segmented viruses in environmental samples, compared to individual birds, and also the fact that assembly could result in chimeric viruses. Assessing the implications of detected variants in such chimeric viruses would be more challenging.

We thank the reviewer for this important observation. We agree that environmental metagenomic surveillance has several limitations compared to traditional host-based sampling, and we have now added a paragraph to the Discussion (lines 1057–1066) to explicitly address these. Most notably, environmental sampling does not allow for direct estimation of virus prevalence within individual animals or flocks, nor can it be used to quantify infection rates at the market level. This limits its utility for certain epidemiological analyses, such as determining attack rates or understanding host-specific infection dynamics. Additionally, environmental samples often represent pooled viral signals from multiple species and individuals, making attribution to a specific host or population impossible without follow-up sampling. Despite these limitations, we emphasize that this approach is not intended to replace conventional sampling, but rather to complement it, offering a broad, scalable, and cost-effective means of early detection, particularly in complex, high-risk settings like live bird markets.

We also agree that for segmented viruses, such as influenza, environmental sampling does not allow for full genome reconstruction or accurate reassortment analysis, and we have now explicitly added this point to the Discussion (lines 1044 to 1047). It's important to mention that there are often individual birds identified with coinfections of H5N1 and H9N2 which also prevents the reliable detection of reassortment events.

We have also revised the manuscript to address the potential for chimeric contigs resulting from assembly artifacts and the resulting challenges in interpreting detected variants (lines 1038 to 1044). The later has been discussed in more detail in response to other comments in this rebuttal.

Section line 311: What is the value of testing mapping of contig-to-contig here? Particularly when contigs are short and sparse, a contig from, e.g., a capsid found in a throat swab, would obviously not map to a contig of an RdRp found in an environmental sample. So is this proportionality more related

to how much of the total genome was recovered in the environmental sample, rather than anything interesting about diversity? It would be useful to clarify this to the reader. If so, perhaps % of genome per type would be a better metric.

We thank the reviewer for their feedback. As mentioned in response to a previous comment we have now greatly expanded on this section (Figure 3).

We first identified the top DIAMOND blastx hit for each poultry contig after taxonomic assignment. Corresponding reference genes were retrieved from GenBank to build a custom database of approximately 4,000 poultry virus genes. A full list of GenBank accessions and associated annotations (genus, species, and gene type) is provided in Supplementary Table 3. All contigs that passed filtering (as previously described) were then mapped to this reference gene set to calculate coverage and percent identity as reported by DIAMOND (Supplementary Table 4). Only DIAMOND hits with a significant e-value threshold greater than $1e-5$ were retained for downstream analysis. Where poultry and ES contigs had significant hits against the same Genbank entry the difference in sequence identity and coverage was reported. If no contig in a given sample aligned to a gene with an e-value $\leq 1e-5$, the gene was considered undetected in that sample. Ultimately this new analysis shows that environmental contigs align to the majority of poultry virus genes with little difference in coverage or percent identity. This analysis is described in the new results section from lines 612 to 753.

We have retained the original analysis being referred to as Supplementary Figures 16 and 17.

In general, there are statistical issues drawing conclusions about the ability of one sample type to capture another, when the number and type of samples is variable per time point. For example, Section line 330: Making such comparisons is challenging given the relatively limited information on sampling. For example, looking at Figure 3, no ducks were swabbed in Takeo in 23rd December 2022. It's perhaps unsurprising then that no H5 viruses were found in bird samples, because H5 viruses are highly duck-adapted rather than chicken adapted. Vice versa, no chicken samples were taken on 21st December – which probably explains the lack of N2s (from chicken-adapted H9N2s). Grouping “poultry swabs” together isn't reasonable, given these different adaptations. Were none of these birds present in the market on these times? I'm sympathetic to the realities of fieldwork, but I do think that these comparisons are misleading/oversimplified as currently presented in the text.

We thank the reviewer for their feedback. In line with one of the previous comments, the number of samples taken at each timepoint and for each individual group was not made clear in the original manuscript. We have now added a table as previously requested to make this clear (Supplementary Table 1). The data being referred to is now Figure 4 in the latest manuscript draft.

At each timepoint irrespective of location there is at least one representative sample from each group. This means that on 23rd of December ducks were sampled (both throat and cloacal) and chickens were also samples on 21st December (throat and cloacal). Figure 3H has caused confusion because it only shows samples that were positive for at least one avian influenza HA (H5, H6, H9) or NA (N1, N2, N6). Thus meaning that H5 was detected in the environment on 23rd of December but not in ducks or other poultry swabs. We have updated this heatmap (Figure 4C) to show all samples across all timepoints.

We understand that different viruses are better adapted to different poultry hosts and thus individual comparisons are incredibly important and informative. As previously mentioned any figure which shows a summary of overall poultry versus ES (such as Figure 1A) is now also broken down into individual comparisons relative to poultry species and sample type and presented as Supplementary Figures.

Line 392: “Given that de novo...”. There's little data presented on whether the de novo assembled sequences do accurately represent those detected in poultry swabs. Whilst we see that they indeed identify the same viruses, the accuracy of the genome information below the broader clade level is not shown. For example, in Figure 3 it's unclear which are new environmental and which are new poultry swab data, and how similar these are. There are some quite long branches in some of the new data, and I would feel somewhat reassured if these weren't in the environmental samples because of the risk of de novo assembly producing strange chimeras. It would have been useful to have some artificial

pools here, e.g., comparing results from sequencing fake “environmental samples” made of pooled swabs, to those obtained from sequencing individual swabs.

We thank the reviewer for their feedback. This has been addressed in relation to several comments already. Figure 3 and associated text at lines 612 to 753 now shows that the majority of environmental contigs align to the same gene as poultry contigs with little difference in sequence identity or coverage.

Regarding the influenza phylogenetic trees that were retained in the latest draft of the manuscript. Branch support was assessed with 1,000 bootstrap replicates, and only branches showing $\geq 70\%$ support are displayed. The extended branches in the phylogenies were attributable to very short contigs from both environmental samples and poultry swab sequences whose limited length can distort tree topology (as you have previously suggested). After applying a $\geq 70\%$ bootstrap filter, these few uninformative tips were removed. Nevertheless, environmental sampling still matches or outperforms poultry swabs in virus detection. Its important to mention that some short sequences still remain in the tree and have been kept as long as they pass the filters described.

Minor comments:

The title suggests that air sampling is sufficient to capture the diversity of zoonotic viruses circulating in the market. Yet there is little clarity provided that any pathogen detected here is zoonotic, with the (implied) exception of avian influenza virus. It would be useful to have some added discussion of the diversity of the zoonotic viruses observed to justify this title.

We thank the reviewer for their comments. We have now updated the title to more accurately reflect the findings if this paper.

Line 24: It is wrong that environmental surveillance has not been used for zoonotic pathogens. The abstract itself defines ES as including air and cage swabs, etc. There is extensive evidence of the use of environmental sample types being used to detect zoonotic pathogens including influenza since at least the 1970s, e.g., see <https://doi.org/10.1111/tbed.13633>. This is recognised by the authors at line 77.

We thank the reviewer for their feedback. We have now altered this statement at line 28.

Line 108: Given that the markets are named, it could be useful to see the point locations on Fig S1.

We have included a new map for Supplementary Figure 1 which shows the specific locations in each province.

Line 120: Additional details should be provided to allow reproducibility, e.g., the type of swab, type of VTM, length of vortex etc. The Direct-zol RNA Miniprep manufacturer’s protocol provides various different conditions for different sample types, and it would be useful to understand which was followed for each sample type, as I imagine that this is different for the pooled swab and the liquid samples. How the samples were pooled into libraries and how many reads per sample and sample type would also be important in the Supplementary Information. The number of PCR cycles in the Twist protocol would be helpful too.

We thank the reviewer for their helpful suggestions. We have now updated the methods section accordingly (lines 166 to 178 and lines 181 to 182) to include additional details of the experimental protocol used. Briefly, samples were collected using polyester swabs and placed into viral transport medium (VTM) containing 2.95% tryptose phosphate broth, 145 mM NaCl, 5% gelatin, 54 mM Amphotericin B, 106 U/L penicillin-streptomycin, and 80 mg/L gentamicin (Sigma-Aldrich, Steinheim, Germany). Samples were vortexed for 15–30 seconds before proceeding directly to RNA extraction using the Direct-zol RNA Miniprep Kit. We used 200 μ L of sample per extraction. Swab samples were pooled by type and species: 10 samples per pool for oropharyngeal and cloacal swabs from chickens and ducks, 5 samples per pool for cage swabs, and 2–5 samples per pool for water samples. Library

preparation included DNA fragmentation at 37 °C for 5 minutes, followed by 10 cycles of PCR amplification, and 22 cycles for post-capture PCR amplification.

Line 178: the R package SourceTracker will not be familiar to all readers, and it would be useful to add some brief details of the approach of this package.

We thank the reviewer for this suggestion. We have now added a brief description of the SourceTracker method to the revised manuscript at lines 277 to 281. Briefly, SourceTracker is a Bayesian approach that estimates the proportional contribution of potential source environments to a given sink sample, based on community composition. The model uses Gibbs sampling to infer the probability distributions of sources contributing to each sink, allowing for the inclusion of an “unknown” source category to account for uncharacterized variation. A reference to the original publication and the package repository has also been included.

Line 204: Why was no bootstrapping or other estimates of nodal support done?

We have now performed 1000 bootstrap replicates for all phylogenetic trees, and only parent branches achieving support values of $\geq 70\%$ are shown in Figure 4 (and associated Supplementary Figures 42-43).

Data availability: Will the consensus sequences be shared individually, at least for those from the individual hosts? Or is this just SRA? I’m glad these data will be shared, but I can’t evaluate how appropriately this has been done without access to the project or the Github.

The raw FASTQ reads will be made publicly available under accession number PRJEB83776. The GitHub is live and the working data is now available there also. It’s important to note that no consensus sequences (only assemblies) were generated in this study.

Line 221: were any novel viruses found? Twist approaches would be able to detect novel viruses, at least for some families. What is used to define whether a virus is “found”, e.g., what if only a short contig of 200bp is present, or not all genes? This is especially important thinking about retroviruses, given the possible similarity to host DNA.

We thank the reviewer. We are aware of TWIST capability to identify novel viruses however all assembled contigs successfully mapped to the nr database with DIAMOND blastx. Thus we could not detect any potential novel viruses.

A virus was considered detected if it had an assembled contig of any length with a substantial number of reads mapping to it, defined as exceeding 0.1% of the total reads in the sample. This provides sufficient phylogenetic signal while minimising false positives. Yes even if the contig is short but still meets the threshold it is included. In addition not all genes from a virus had to be recovered for that virus to be included. Regarding the latter, our aim was to determine whether environmental samples (ES) recover the same viral taxa and genes as poultry swabs not to assemble complete genomes. Metagenomic studies routinely accept partial genomes especially for segmented or highly variable RNA viruses and none require every gene to be present for a detection to be reported.

If a high risk virus X is found at an LBM in Cambodia but its only a 200bp contig or not all genes for that virus are found should this just be disregarded? The main goal of virus surveillance is risk mitigation so it shouldn’t matter what the length of the contig is or if all genes for a given virus can be found. Of course we completely understand concerns about false positives and we have already filtered our dataset to deal with this potential issue.

From a risk mitigation perspective, the appropriate next step would not be dismissal but rather a structured follow-up investigation, including classical molecular diagnostics (e.g., targeted PCR), confirmatory sequencing, and potentially additional environmental and host sampling. These secondary efforts would help validate the signal, assess whether the virus is replicating and circulating, and

determine the appropriate level of response. Excluding findings solely based on contig length risks missing early indicators of emerging threats. In surveillance for pandemic-potential viruses, even weak signals, if well-controlled, can be critical to catching the threat before it escalates.

Line 231: is gull coronavirus definitely proven as respiratory, not fecal-oral?

We thank the reviewer for raising this important point. While avian coronaviruses, including those found in gulls, have been primarily associated with respiratory and enteric tracts, definitive experimental evidence establishing the exclusive transmission route (respiratory vs. fecal-oral) for gull coronavirus remains limited. We have now removed this sentence.

Line 248: I don't understand this conclusion. Figure 1C shows the number of viruses detected, but not the diversity. So it's not clear whether more sample types would be necessary, or, e.g., if wash water contains only a subset of air sample data.

We thank the reviewer for these comments. Overall we agree that Figure 1C in the original manuscript draft was not clear and failed to adequately support the statement being referred to here.

To address this concern we have included a new Supplementary Figure 11 which is a stacked bar plot which breaks down viruses detected in multiple (≥ 2) environmental sample types (shown at the base of each bar) and those unique to a single environmental sample type (stacked on top). Importantly, this analysis only focuses on the direct recapture of viruses found in poultry at each specific timepoint. This provides a clearer picture of how virus detection is distributed across sample types. It can be seen that for the majority of visits to the LBM that the total number of species recaptured for each poultry swab is increased by detections unique to specific environmental samples at that given time. In this current study each environmental sample type is shown to contribute to this at least once by increasing the number of viruses recaptured. Thus in order to recapture as many viruses as possible a variety of environmental sample types should be used.

In addition, when considering the total number of poultry viruses detected (not just those recaptured) we show in Figure 1B that air from the slaughter and holding areas, as well as cage swabs have a higher alpha-diversity than other groups. In Figure 2D we show that these three ES types are the shortest distance from the median centroid of chicken cloacal swabs. While in Figure 2C cage swabs are shown as the group with the second shortest distance from oropharyngeal swabs in chickens. Together these findings based on both alpha-diversity and beta-diversity analysis indicate that air (slaughter and holding area) and cage swabs enhance poultry virus surveillance and thus must be included. In Figure 2E-F we see that wash water was the shortest distance from oropharyngeal and cloacal swabs in ducks suggesting it performed best at detecting viruses from this host. This is important as in Figure 1 and associated Supplementary we often see wash water detecting less viruses than other ES types. This tells a story of quality over quantity while wash water is less diverse (alpha-diversity) it shared a higher proportion of their fewer viruses with duck samples highlighting their importance. This analysis also captures temporal consistency, as it incorporates data from all twelve sampling visits to the live bird market. The same can be said for drinking water which was the shortest distance from oropharyngeal swabs in chickens (Figure 2C).

Five out of six ES types in this study contributed meaningfully to poultry virus detection, supporting the inclusion of diverse sample types to maximize surveillance performance. Based on these findings and comments from other reviewers we have now described the optimal sampling strategy in the discussion on lines 1048 to 1057.

Line 277 to 280: I don't understand why this approach was used rather than something simpler, please clarify to the reader. Line 283 "tended to have more negative correlations", is also unclear.

We are not entirely certain what is meant by a "simpler" analysis in this context, but we appreciate the opportunity to clarify our approach. In line with comments from other reviewers and the large number of changes made since the first draft, this analysis is now presented as Figure 2G-J and the associated

text is on lines 553 to 571. We have made significant changes to the text and we believe the results described are now with clearer.

Ultimately a simple correlation analysis calculated between two types of data is all that has been done here. On one hand we have the distance of each sample from each poultry median centroid (four separate distance measures) as is presented in Figure 2C-F. On the other hand we have virus abundance in RPM. The resulting correlation matrix has the four distance measures as rows and the virus species as columns. Based on analysis in Figure 1 we generated a list of taxa which were successfully recaptured and those which failed to be detected by ES. So taxa are split into two groups. Thus in Figure 2G-J the spearman correlation coefficient obtained for each virus is represented as a datapoint.

Negative correlation indicates that two values are moving in the opposite direction. So using this approach if any one virus has a negative correlation it means that as the distance to the median centroid becomes shorter (low value) the abundance of the virus increases (high value). This scenario is a positive association between the virus and the group represented by the median centroid. Conversely, for the same virus, if the distance from the median centroid increases (high value) the abundance of this virus decreases (low value). This is also a positive association between the virus and the median centroid.

On the other hand, given that positive correlation indicates two values are moving in the same direction. If one virus has a positive correlation it means that the distance to the median centroid becomes longer (high value) as the abundance of the virus increases (high value). This is a negative association between the virus and the median centroid. On the other hand if the distance from the median centroid become shorter (low value) the abundance of the virus decreases (low value). This is also a negative association.

Thus using our approach, negative correlations are positive associations while positive correlations and negative associations.

Focusing now on Figure 2H we see that viruses successfully “recaptured” by ES have a significantly lower spearman distance than viruses “undetected”. Just by visualising the boxplot we see that for “recaptured” taxa the 75th percentile (Q3) is around 0. Thus the majority of viruses in this group (“recaptured”) have a negative spearman correlation coefficient with the distance from the median centroid of chicken cloacal. Thus for the majority of recaptured viruses the shorter the distance from the median centroid of chicken cloacal the higher their abundance (positive association). For undetected taxa the opposite is true as the 0 line is somewhere between the 25th and 50th percentile. This means the majority of viruses in this group have positive correlations. Thus viruses undetected by ES have a negative association with chicken cloacal swabs. Clearly we can see the opposite is true for ducks and again this finding is statistically supported.

We believe that this beta-diversity analysis combined with median centroid testing and correlations is very informative approach for identifying associations between ES performance and poultry host/sample type. We hope this clarification resolves any confusion and have revised the relevant sections in the manuscript to explain this more clearly.

Line 306 – “performs better at detecting chicken viruses” – perhaps this conclusion could be phrased as “performs better at detecting viruses from the chickens, which were also the most abundant host in this market”, or equivalent. Whilst I note your Discussion point about this, I think it’s useful to add this note here or at least a link to the Discussion. Adding information on the total estimated relative numbers/proportion of individuals per species in each market would be critical here.

We thank the reviewer for their feedback. We have now updated the text at lines 604 to 606. We have also now included the estimated numbers in the discussion (lines 1020 to 1021). This is based on an FAO report which is cited in the discussion. Between 10 and 100 chickens are slaughtered per day while between 1-10 ducks are.

Figure 3H doesn't appear to have any text associated with it in the legend. Should it also say N6, not H6?

The figure legend for panel 3H (now Figure 4H) has been updated for clarity at lines 1548 to 1554. Additionally, we have corrected a labelling error, changing "H6" to "N6" as originally intended.

Line 403: Correct to "Gamma"

This analysis has now been removed.

Response to reviewers' comments

Please note that the line numbers mentioned in our response refer to the revised manuscript with tracked changes (all markup). Please see our responses in blue, and the line numbering follows the clean version.

REVIEWER COMMENTS

Reviewer #1 (Remarks to the Author):

The revisions fully address all of my concerns.

A minor note on some of the supplemental figure labels. Many of the labels from figures with corresponding tables of statistical results have this at the end:

"P values for all pairwise comparisons can be found in Supplementary File X-Y. P values are annotated as follows: P < 0.05 *; P < 0.01 **; P < 0.001****". The actual table numbers are missing and the asterisks are missing from the figures.

We thank the reviewer for their feedback. We have now updated all Supplementary Figure legends to include the correct references to the corresponding Supplementary Tables.

To clarify to the editor, given the large number of pairwise comparisons in some figures, we selectively included asterisks to highlight only the most relevant or representative comparisons directly on the plots. The full list of P-values and significance levels for all pairwise tests is provided in the corresponding Supplementary Tables as indicated in the updated figure captions.

Reviewer #3 (Remarks to the Author):

Thank you to the authors for thoroughly addressing most of my comments. The additional detail, new analyses and new figures have really improved the quality of this paper, and the ability of readers to fully understand the data being presented. I have only a few outstanding comments:

- Thanks for adding Figure S2. It is useful to see that no poultry-virus reads were identified in the negative controls. However, please add detail comment about what the negative control sample was (for example, was this a water control, processed from the point of extraction alongside all other samples analysed here? Or a completely separate run? Or something else?). The choice of negative control has important implications for interpretation. If the negative control was not processed alongside these samples, that is a significant weakness. (Please note that I disagree with the author's comment in the rebuttal that "sterile" air filters are unlikely to be informative as a negative control. Sterile implies inactivation of pathogens, and does not imply RNA/DNA free status. I agree that it's unlikely that filters would contain poultry pathogens, but use of a blank negative control that uses all of the plasticware and reagents and is run alongside real samples is best-practice for metagenomic studies including facilitating subsequent reanalysis of SRA data where people may not just be interested in poultry viruses. But I agree that it's not critical here - it's just critical to state what the actual choice of negative control was as noted above).

We thank the reviewer for their feedback. We have updated the Methods section (lines 180–184) accordingly. Nuclease-free water was included as a negative control and processed alongside both poultry and environmental samples from the point of extraction through to the library preparation. In addition, to be clear, negative controls were then included on the same Illumina sequencing run as the actual samples themselves.

- Thanks for adding information on the number of samples per time point. Please confirm in the text that samples from individual birds each came from a different stall (i.e., the birds were randomly sampled from all birds at the market). This is important because birds from the same stall/cage are

more likely to share the same viruses, compared to environmental samples from different stalls. This sampling design is valuable for the reader to understand, especially when many field studies do not do this as it's harder to sample in this way in the field.

We agree that this clarification is important. Chickens and ducks were indeed randomly sampled from across the entire market rather than from individual cages/stalls. We have now updated the manuscript text at lines 125–126 to specify this.

- To be clear to the editor and the authors, my question about “were any novel viruses found” is not intended to “dismiss” viruses where only short contigs are found as the authors imply. I fully agree that short contigs can be informative. However, it is important to provide clarity to the reader about what was counted as a “detection” here and whether any effort was made to discriminate between known and novel viruses. This clarity is necessary when interpreting the manuscript for readers who may be interested in virus diversity in these settings. The rebuttal now states that they do not attempt to detect novel viruses. To aid the reader’s understanding, it should be clearly stated somewhere in the manuscript - perhaps the Discussion? - that just because a virus has a close hit in nr using DIAMOND, it does not mean that the contig is 100% conclusively that virus. This is a recognised problem with metagenomic studies (I am reminded of <https://doi.org/10.1016/j.cels.2015.07.008!>).

Similarly, I would recommend removing the new text in the concluding paragraphs that states that “environmental samples enable detection of both known and novel viruses across multiple hosts and sample types”.

We thank the reviewer for this helpful clarification. We agree that, although all contigs showed significant similarity to known viruses in the nr database, sequence similarity does not always guarantee complete certainty about viral identity, particularly for short or partial contigs. We have clarified this point in the Discussion (lines 666–668) to ensure readers understand that taxonomic assignments are based on best available similarity matches, and we have removed the statement referring to the detection of “novel” viruses to avoid any potential confusion.

- Thanks for stating that you now use bootstrapping on trees. However, I don’t understand how removal of parental branches with bootstraps <70 was done on Figure 4 and S42/S43 – do NONE of the internal branches have bootstraps <70? I can understand how pruning of external branches and corresponding samples could have been done, but it seems unlikely that removal of internal branches with bootstraps <70 would be possible without leaving a partially unconnected tree. Please check, and make the process of this filtering clear to the reader with a caveat somewhere about the potential issues that can be caused by phylogenetic estimation on mixed length sequences where aligned regions may come from different genome sections in different viruses.

We thank the reviewer for this helpful comment and the opportunity to clarify our approach. Following bootstrap resampling, we filtered the phylogeny by removing only terminal branches (tips) whose immediate parental node had bootstrap support below 70. This was performed in R using the ape and ggtree packages. For clarity, the ape::drop.tip() function automatically prunes these low-support tips and reconnects the remaining nodes, preserving a single, fully connected tree. We have now clarified this procedure in the Methods section (lines 290–295) and note the caveats associated with phylogenetic estimation on mixed-length alignments where some regions may represent different genome sections (lines 674 and 677).

- Figure S16: the rebuttal says that “this heatmap being referenced shows abundances as ranked RPM which is not the same as raw RPM”. Thanks for adding the information on rank scaling to the methods. Please can you modify the legend slightly to make this clearer to the reader? “Heatmap showing the abundance of non-poultry viruses detected, represented as reads per million” implies raw to me, even though Rank(RPM) is on the figure.

We have now updated the Figure Legend for S16 as requested.

- There are a lots of typos generally in the newly added sections (e.g., “racptured” in S5 legend, “environmental” multiple times in S11). Please check.

We have now revised all Supplementary Figures and Figure legends to fix these typos.

- In the phylogenies:

o it’s almost impossible to see out the tip shape in some of the phylogenies where the tip shapes are overlapping. Perhaps the trees could also be provided as separate nexus/newick formats files, to help with this?

o The tip shape legends in all trees in the main text and supplementary are hard to understand (e.g., the Figure 4 legend doesn’t mention the square tip. Are the poultry samples not generated here as “new sequences”? If not new, what differentiates these “poultry” from “reference”?

o Please make all the trees use the same tip shape for the same tip type. For example, it really confused me for a long time that in 4A, triangles are "reference" but in 4B, triangles are "new sequence". This makes it hard to read, but also inconsistent with the legend text.

o I don't really understand the red arrow or what they mean. They seem to point to quite vague points on the tree, whereas generally I think that highlighting a small clade is the intention. Consider, perhaps, changing the tip opacity for the samples from these markets instead (i.e., more transparent for samples from the reference dataset might work well, and more clearly show the clustering of different sample types from this study).

We thank the reviewer for these valuable suggestions regarding the phylogenetic trees. The tree files used to plot Figure 4 are already available on the GitHub along with other data analysis files and code. To improve the clarity and consistency of the phylogenies, we have updated all trees so that global tips now includes a thin black outline to enhance definition and distinguish overlapping symbols. New sequences from this current study have been given a red tip outline. The legends have been corrected and standardized across all figures, squares now represent poultry sequences, circles represent environmental sequences, and triangles represent reference sequences downloaded from GISAID or NCBI. To further improve visibility, poultry and environmental tips have been made slightly larger than reference tips, and the opacity has been adjusted such that reference sequences appear lighter while poultry and environmental tips appear darker, as suggested. As you mentioned, the red arrows were originally intended to draw attention to the location of poultry and environmental sequences within each tree; we have now removed these. We agree that these modifications substantially improve the readability, consistency, and interpretability of the phylogenies.